# Evolutionarily distant I domains can functionally replace the essential ligand-binding domain of *Plasmodium* TRAP

**Dennis Klug[1,2]\*, Sarah Goellner[1,3], Jessica Kehrer[1], Julia Sattler[1], Léanne Strauss[1], Mirko Singer[1,4], Chafen Lu[5], Timothy A Springer[5]\*, Friedrich Frischknecht[1]\***

[1]Integrative Parasitology, Center for Infectious Diseases, Heidelberg University Medical School, Heidelberg, Germany; [2]Université de Strasbourg, CNRS UPR9022, INSERM U963, Institut de Biologie Moléculaire et Cellulaire, Strasbourg, France; [3]Department of Molecular Virology, Heidelberg University Medical School, Heidelberg, Germany; [4]Experimental Parasitology, Faculty of Veterinary Medicine, Ludwig-Maximilians-Universität München, München, Germany; [5]Program in Cellular and Molecular Medicine, Children's Hospital Boston, and Departments of Biological Chemistry and Molecular Pharmacology and of Medicine, Harvard Medical School, Boston, United States

**\*For correspondence:**
dennis.klug@sciencebridge.net
(DK);
springer@crystal.harvard.edu
(TAS);
freddy.frischknecht@med.uni-heidelberg.de (FF)

**Competing interests:** The authors declare that no competing interests exist.

**Abstract** Inserted (I) domains function as ligand-binding domains in adhesins that support cell adhesion and migration in many eukaryotic phyla. These adhesins include integrin αβ heterodimers in metazoans and single subunit transmembrane proteins in apicomplexans such as TRAP in *Plasmodium* and MIC2 in *Toxoplasma*. Here we show that the I domain of TRAP is essential for sporozoite gliding motility, mosquito salivary gland invasion and mouse infection. Its replacement with the I domain from Toxoplasma MIC2 fully restores tissue invasion and parasite transmission, while replacement with the aX I domain from human integrins still partially restores liver infection. Mutations around the ligand binding site allowed salivary gland invasion but led to inefficient transmission to the rodent host. These results suggest that apicomplexan parasites appropriated polyspecific I domains in part for their ability to engage with multiple ligands and to provide traction for emigration into diverse organs in distant phyla.

## Introduction

Domains with similar overall structures, initially described in von Willebrand factor A (VWA domains), are found in cell-surface proteins including integrins, extracellular matrix, and complement components, and mediate a diversity of functions including cell adhesion, migration, and signaling (*Whittaker and Hynes, 2002*). Here, we study a subset of VWA domains termed I domains because they are inserted in other domains in integrins. I domains differ from VWA domains in the position of their ligand binding sites and in the presence of a metal ion-dependent adhesion site (MIDAS) at the center of their ligand binding site (*Liddington, 2014*). Within integrins, I domains switch between closed and open states coordinately with conformational change in neighboring domains. This switch from closed to open conformation in the I domain alters the ligand-binding site around the MIDAS and increases affinity for ligand by ~1,000 fold (*Schürpf and Springer, 2011*).

I domains are key modules in adhesins employed by apicomplexan pathogens. I domain-containing, membrane-spanning surface glycoproteins have been shown to be essential for tissue traversal and cell invasion by *Toxoplasma gondii* and *Plasmodium spp.* and are present in all known apicomplexans (*Sultan et al., 1997*; *Morahan et al., 2009*). In *Plasmodium (P.)*, a protein named CTRP

**eLife digest** Malaria is an infectious disease caused by single-celled parasites known as *Plasmodium*. Humans and other animals with backbones – such as birds, reptiles and rodents – can become hosts for these parasites if an infected female mosquito feeds on their blood. Likewise, healthy mosquitoes can in turn become infected with *Plasmodium* if they feed on the blood of an infected animal.

To complete their life cycle, *Plasmodium* parasites within a mosquito must become spore-like cells called sporozoites. These sporozoites are highly mobile and can get into the mosquitoes' salivary glands, meaning they can be passed on to a new host when the insect feeds. During a mosquito bite the sporozoites are spat into the skin of the potential host, where they then need to migrate rapidly to enter the bloodstream. Once in the blood, the sporozoites can then get into liver cells and begin a new infection. One protein called TRAP, which is found on the surface of the sporozoites, is important for their migration and the infection of the salivary glands or liver. Yet it was not known how this happens at the level of the individual proteins involved.

Klug et al. have now tested how a part of the TRAP protein, called the I domain, contributes to the infection process. In the experiments, the I domain of TRAP was deleted which showed that the sporozoites need this domain to be able to move around and get into the host tissues. Without the I domain the sporozoites were stuck and could not successfully infect either the mosquitoes, the livers of mice, or human liver cells grown in the laboratory. Klug et al. then replaced the *Plasmodium* I domain of TRAP with the I domain from a distantly related parasite called *Toxoplasma gondii*, which causes a condition known as toxoplasmosis. The I domain from *Toxoplasma* allowed the *Plasmodium* parasites to infect the host tissues again. This observation was unexpected because *Toxoplasma* and *Plasmodium* parasites have evolved separately over the last 800 million years and *Toxoplasma* does not infect insects.

These findings suggest that the I domain of TRAP evolved to bind several other proteins in different tissues and hosts. Future studies will investigate which other parasite proteins TRAP works with to guide sporozoites to the salivary glands or liver. Knowledge of how these proteins act together may lead to new approaches for treating or preventing malaria. For example, some treatments could stop sporozoites from entering liver cells.

containing six I domains is required for invasion of the mosquito midgut by the ookinete (*Ramakrishnan et al., 2011*; *Dessens et al., 1999*). Once the ookinete crosses the midgut epithelium it forms an oocyst wherein it differentiates into hundreds of sporozoites (*Frischknecht and Matuschewski, 2017*). Sporozoites use proteases (*Aly and Matuschewski, 2005*) and active motility (*Klug and Frischknecht, 2017*) to egress from the oocyst into the hemolymph and subsequently enter the salivary glands from where they can be transmitted back to a vertebrate host. Once deposited in the skin during a blood meal by an infected mosquito, sporozoites migrate rapidly to find and enter blood vessels (*Amino et al., 2006*). Within the blood stream parasites are passively transported to the liver, where they infect hepatocytes and develop into liver stages (*Douglas et al., 2015*). The subsequent blood stages cause the typical symptoms of malaria by triggering a massive immune response, clogging capillaries and lysing red blood cells (*Cowman et al., 2016*).

Sporozoites express two adhesins with I domains, TRAP (thrombospondin related anonymous protein) and TLP (TRAP-like protein). These proteins are stored in secretory vesicles called micronemes, at the apical end of the highly polarized sporozoite (*Tomley and Soldati, 2001*). After fusion of micronemes with the plasma membrane, TRAP and TLP decorate the surface of the sporozoite and form a bridge between extracellular ligands and the membrane-subtending actin-myosin motor that drives gliding motility and invasion (*Heintzelman, 2015*; *Frischknecht and Matuschewski, 2017*). Deletion of *tlp* causes only a mild phenotype in tissue traversal while deletion of *trap* yields sporozoites that cannot move productively in vitro, fail to enter salivary glands, and are unable to infect mice if isolated from mosquitoes and injected intravenously (*Sultan et al., 1997*; *Moreira et al., 2008*; *Hellmann et al., 2013*; *Quadt et al., 2016*). Mutations of amino acids within the MIDAS motif of the single I domain in TRAP decreased the capacity of sporozoites to enter salivary glands and liver cells as well as to infect mice (*Wengelnik et al., 1999*; *Matuschewski et al.,*

*2002*). However, these mutant sporozoites were still able to migrate in vitro. This suggests that the MIDAS is important for ligand binding but not for productive motility.

Crystal structures of the N-terminal portion of TRAP in *Plasmodium spp.* and a TRAP orthologue in *Toxoplasma gondii,* the micronemal protein 2 (MIC2), revealed the I domain in both open and closed conformations in association with a thrombospondin type-I repeat domain (*Song et al., 2012*; *Song and Springer, 2014*; *Figure 1*). The apicomplexan I domains structurally resemble I domains found in integrin α-subunits (αI domains) much more than I domains in integrin β-subunits (βI domains). Between the closed and open states of both apicomplexan I domains and integrin αI domains, the $Mg^{2+}$ ion at the MIDAS similarly moves ~2 Å closer to one coordinating sidechain and away from another. This movement is linked to essentially identical pistoning of the C-terminal, α7-

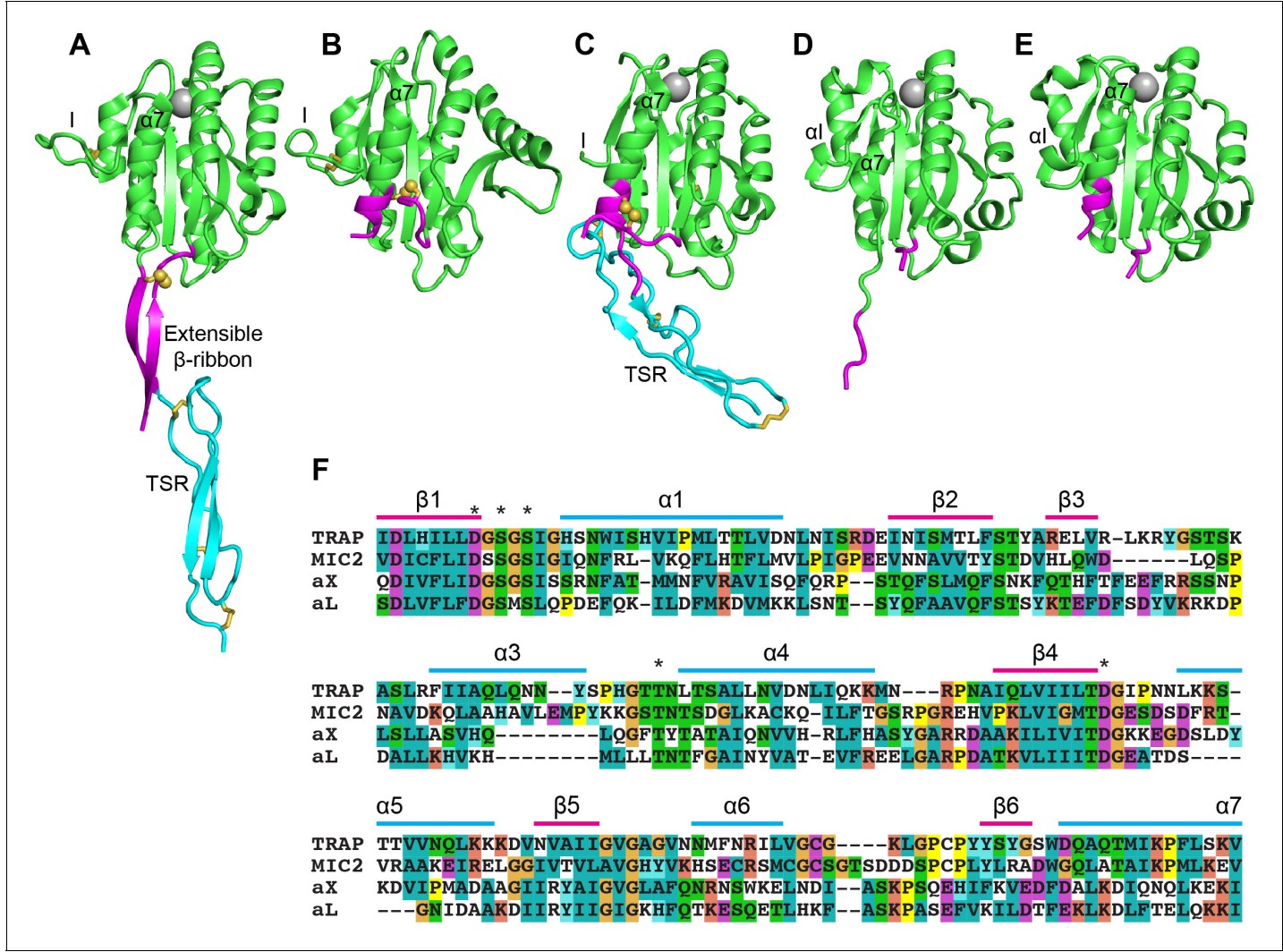

**Figure 1.** Structural features of TRAP, MIC2, and integrin αI domains. (A–E) Cartoon ribbon diagrams. (A), open TRAP (Protein Databank ID (PDB)) 4HQL; (B), closed TRAP, PDB 4HQF; (C), closed MIC2, PDB 4OKR chain B; (D), open αX αI domain, PDB 4NEH; (E), closed αX αI domain, PDB 5ES4. Green: the portion of the I domains exchanged between TRAP and other I domains; cyan: the TSR domain; magenta: the portion in between the I domain and TSR in TRAP and MIC2 which includes the extensible β-ribbon. The comparable regions in integrin αI domains are also colored magenta and emphasize how the α7-helix reshapes similarly to that of apicomplexan I domains. The TSR domain of closed TRAP (B) is not shown because it was disordered in crystals; it likely is positioned similarly in the closed conformation as the TSR domain of closed MIC2 (C). The MIDAS $Mg^{2+}$ ion is shown as a silver sphere and is present in all I domains. It is not shown in closed TRAP (B) because a lattice contact disrupted the conformation around the MIDAS. Disulfide bonds are shown as yellow sticks; for emphasis, the cysteine sulfur atoms of the extensible I domains are shown in identical orientations after superimposition. (F) The portions of the four I domains that were exchanged were aligned by sequence and structure (*Song et al., 2012*; *Song and Springer, 2014*). Secondary structure elements are labeled and MIDAS residues are asterisked above the alignment.

helix toward the 'bottom' of the domain (*Figure 1A* compared to 1B and *Figure 1D* compared to 1E). The distance pistoned is equivalent to two turns of an α-helix. Uniquely in the apicomplexan I domains, a segment N-terminal to the I domain is disulfide-linked to the last helical turn of the α7-helix in its closed conformation (*Figure 1A–C*). As this segment pistons out of contact with the remainder of the I domain in the open conformation, the last two turns of the α7-helix with its cysteine unwind, the N-terminal segment with its cysteine moves in a similar direction, and these segments reshape to form a β-ribbon (*Figure 1A*). Because of close structural homology between human integrin αI- and apicomplexan adhesin I domains in the regions shown in green in *Figure 1A–E*, we were able to engineer exchanges between them in this study (*Figure 1F*). *P. berghei* parasites expressing TRAP without an I domain phenocopy *trap(-)* parasites and show that the I domain is essential for motility and invasion. Parasites expressing TRAP with the I domain of MIC2 from the related apicomplexan *Toxoplasma gondii*, show rescued motility and invasion. By reversing the charges of amino acid residues around the MIDAS motif we could partially uncouple the function of TRAP during salivary gland invasion and rodent infection. Our results show that I domains have the capacity to be poly-specific and permit TRAP to function as an adhesin in both vertebrate and arthropod hosts.

## Results

### The I domain of TRAP is crucial for *Plasmodium* transmission

The importance of the I domain for TRAP function was tested using the *trapΔI* parasite line (*Figure 2A*, *Figure 2—figure supplement 1*). In several independent experiments in which mosquitoes fed on infected mice, only few *trapΔI* or *trap(-)* sporozoites could be observed within the mosquito salivary glands, whereas for *trap* wild type ~10,000 sporozoites were observed per mosquito (*Figure 2B*, *Table 1*). Thus, *trapΔI* sporozoites are similarly impaired in salivary gland invasion as *trap(-)* sporozoites. To determine whether mutant parasites retained the ability to migrate steadily on microscope slides, that is to glide in circles (*Vanderberg, 1974*), sporozoites were isolated from hemolymph and activated by addition of 3% bovine serum albumin (BSA). Hemolymph sporozoites, like midgut sporozoites, were present in all mutants studied here (*Table 1*). Sporozoites were defined as gliding and productively motile if they were able to complete at least one circle within 3 or 5 min, depending on the experiment. Sporozoites exhibiting other types of motion were classified as unproductively motile, while sporozoites that were attached but were not moving or were not attached were classified as non-motile (*Figure 2—figure supplement 2*; *Münter et al., 2009*). Hemolymph sporozoites were ~19% productively motile in wild type while none were motile in *trapΔI* (*Figure 2C*) and *trap(-)* mutants (*Münter et al., 2009*; *Hegge et al., 2010*; *Figure 2C*). These results show that the I domain is required for productive motility.

Infectivity of mutant sporozoites was tested by exposing naive mice to infected mosquitoes or by intravenously injecting sporozoites obtained from the midgut, hemolymph or even salivary glands (*Table 1*). Upon infection with wild type sporozoites, the first blood stage parasites were visible as expected after 3 to 8 days; in contrast, no infections could be observed for *trapΔI* and *trap(-)* parasites (*Figure 2—figure supplement 3*, *Table 2*, *Table 3*). Immunofluorescence assays on midgut sporozoites using an antibody recognizing the repeat region of TRAP showed specific fluorescence in most sporozoites at one end with no recognizable difference between *trapΔI* and wild type sporozoites. This suggests that the mutated TRAP is also localized in the secretory micronemes. In contrast, TRAP-specific fluorescence was absent in *trap(-)* sporozoites. TRAP could also be observed on the surface of unpermeabilized *trapΔI* sporozoites indicating that micronemal secretion is not abolished in these parasites (*Figure 2D*).

### Structurally conserved I domains of other species partially rescue salivary gland invasion

We next tested whether the lack of productive motility and infectivity as well as the severely impaired salivary gland invasion rate in *trapΔI* parasites could be rescued by replacing the deleted TRAP I domain with an I domain from a foreign species. We selected structurally characterized I domains of MIC2 from *Toxoplasma gondii* and the I domains of the human integrin α-subunits αX (CD11c) and αL (CD11a) (*Figure 1C–E*). Sequence identity among I domains is 36% between αL and

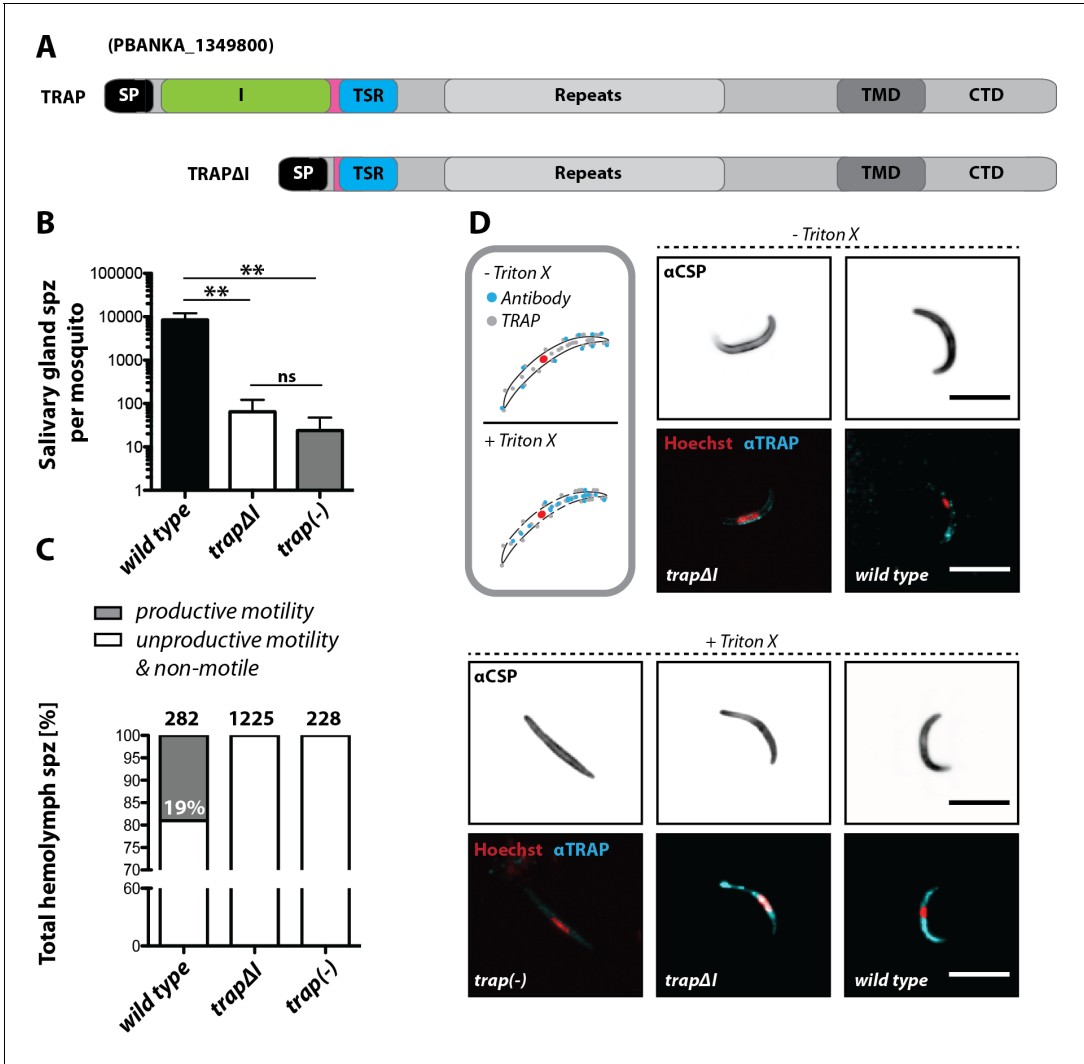

**Figure 2.** The I domain of TRAP is essential for salivary gland invasion and gliding motility of sporozoites. (A) Domain architecture of full-length TRAP and the mutant TRAPΔI lacking the I domain denoting signal peptide (SP), I domain (I, green), thrombospondin type-I repeat (TSR, blue), repeats, transmembrane domain (TMD) and cytoplasmic tail domain (CTD). (B) Sporozoite numbers in the salivary glands of mosquitoes infected with *trapΔI*, *trap(-)* and wild type (*wt*) 14–22 days post infection. Shown is the mean ± SEM of at least seven counts of three different feeding experiments. ***p<0.0001 one-way-ANOVA (Kruskal-Wallis-test). (C) Motility of hemolymph sporozoites isolated 13–16 days post infection. Sporozoites moving in at least one full circle within five minutes were considered to be productively moving while all sporozoites that behaved differently were classified as unproductively moving/non motile. The number of analyzed sporozoites is indicated above each bar. (D) Immunofluorescence images of permeabilized and unpermeabilized *trapΔI*, wild type and *trap(-)* midgut sporozoites using αTRAP-repeat and anti CSP antibodies as well as Hoechst to stain DNA. Note that intracellular TRAP is stained after permeabilization as indicated in the schematic (grey box). Scale bars: 10 µm.

The online version of this article includes the following figure supplement(s) for figure 2:

**Figure supplement 1.** Generation of *trapΔI* and *trap(-)* parasites.
**Figure supplement 2.** Movement patterns exhibited by sporozoites.
**Figure supplement 3.** *trapΔI* sporozoites are not infective to mice.

αX, 18% between these integrins and *P. berghei* TRAP, and 28% between TRAP and MIC2 (*Supplementary file 2*). The I domains from TRAP and αX are basic, with pI values of 9.7 and 8.9, respectively, while those of MIC2 and αL are acidic, with pI values of 6.1 and 5.8, respectively (*Song et al., 2012*; *Song and Springer, 2014*). Furthermore, the αX I domain is poly-specific as shown by binding to multiple glycoproteins and proteolytic fragments as well as heparin (*Vorup-Jensen et al., 2005*; *Vorup-Jensen et al., 2007*), while the αL I domain is highly specific for the

**Table 1.** Absolute sporozoite numbers in midgut (MG), hemolymph (HL) and salivary glands (SG) of all analyzed parasite strains. Sporozoites in the midgut, hemolymph and the salivary glands of infected mosquitoes were counted between day 14 and day 24 post infection of each feeding experiment. Shown is the mean ± SD of all performed experiments per parasite line. Mosquitoes infected with fluorescent parasite lines were pre-selected for fluorescence in the abdomen using a fluorescence binocular microscope while mosquitoes infected with non-fluorescent parasites were not. Hence sporozoite numbers per infected mosquito for fluorescent parasite lines should be higher compared to non-fluorescent lines. n.d. – not determined; bold numbers indicate lower levels of salivary gland invasion.

| Parasite line | No. of MG sporozoites | No. of HL sporozoites | No. of SG sporozoites | SGS pos./total counts | SGS/MGS |
|---|---|---|---|---|---|
| trap(-) | 26,000 (±24.000) | 6000 (±7,000) | **20** | 1/8 | **0.0009** |
| wt | 10,000 (±3,000) | n.d. | 8000 (±7,000) | 4/4 | 0.8 |
| fluo | 108,000 (±70,000) | n.d. | 21,000 (±4,000) | 6/6 | 0.2 |
| TRAP-I | 26,000 (±7,000) | 8,000* | 16,000 (±4,000) | 2/2 | 0.6 |
| MIC2-I | 38,000 (±16,000) | 6,000* | 18,000 (±3,000) | 2/2 | 0.5 |
| αX-I | 35,000 (±13,000) | 4,000* | **210** | 2/2 | **0.006** |
| αL-I | 42,000 (±16,000) | 7,000* | **30** | 1/2 | **0.0006** |
| RevCharge | 70,700 (±12,000) | n.d. | 13,000 (±6,000) | 7/7 | 0.2 |
| trapΔI | 24,000 (±20,000) | 4000 (±5.000) | **60†** | **3/14** | **0.003** |
| TRAP-I fluo | 22,000 (±13,000) | 2000 (±2.000) | 17,000 (±6,000) | 8/8 | 0.8 |
| MIC2-I fluo | 21,000 (±17,000) | 1000 (±700) | 7000 (±4,000) | 7/7 | 0.3 |
| αX-I fluo | 37,000 (±7,000) | 6000 (±2,000) | **120** | 6/6 | **0.003** |
| αL-I fluo | 34,000 (±9,000) | 5000 (±3,000) | **90** | 3/6 | **0.003** |

*hemolymph sporozoites of the non-fluorescent lines TRAP-I, MIC2-I, αX-I and αL-I were only counted once.

†in one infection 800 SG sporozoites could be counted. Intravenous injection of 5000 salivary gland sporozoites into each of four mice did not lead to infection.

ligand intercellular adhesion molecule (ICAM-1) and its homologues ICAM-2, ICAM-3, and ICAM-5 (**Grakoui et al., 1999**).

The *fluo* line, with eGFP constitutively expressed in all parasite stages and mCherry specifically expressed in sporozoites, was used to generate some of the respective transgenic parasite lines to simplify analysis of TRAP I domain replacements throughout the parasite life cycle (**Bane et al., 2016**). Parasite lines expressing either codon modified wild type TRAP (TRAP-I) or codon modified TRAP with foreign I domains (MIC2-I, αX-I or αL-I) replacing the *P. berghei* TRAP I domain (**Supplementary file 2**) were generated with both *fluo* and *non-fluo trap(-)* parasite lines by homologous recombination (**Figure 3A** and **Figure 3—figure supplement 1**). TRAP shows the same localization in *MIC2-I*, *αX-I* and *αL-I* midgut sporozoites as in *TRAP-I* midgut sporozoites (**Figure 3B**). Western blots showed similar expression levels of TRAP in hemolymph sporozoites of *TRAP-I*, *trapΔI*, *αX-I* and *αL-I* while no TRAP could be detected for *trap(-)* (**Figure 3C**). Similar localization and expression of TRAP was also observed for *MIC2-I* and *TRAP-I* salivary gland sporozoites (**Figure 3D, E**). *MIC2-I fluo* and *non-fluo* lines entered salivary glands at similar rates as wild type sporozoites (**Figure 3F,G,H**, **Table 1**). *αX-I* and *αL-I* sporozoites were capable of invading the salivary glands, albeit at very low rates (**Table 1** and **Figure 3F and G**, note the log scale of the y axis). The numbers

**Table 2.** Determination of transmission efficacy in vivo.

Transmission efficacy of *trap(-)*, *trapΔI*, *TRAP-I*, *MIC2-I*, *αX-I*, *αL-I* and *RevCharge* as well as the wild type (*wt*) reference line. Prepatency is the time between sporozoite infection and the first observance of blood stages and is given as the mean of all positive mice of the respective experiment(s). C57BL/6 mice were either injected intravenously (i.v.) with 10,000 salivary gland sporozoites (SGS) or 10,000 hemolymph sporozoites (HLS) or exposed to 10 mosquitoes that received an infected blood meal. Note that mosquitoes infected with fluorescent parasite lines were pre-selected while mosquitoes infected with non-fluorescent parasites were not pre-selected but dissected afterwards to ensure that all mice were bitten by at least one infected mosquito. For strains with strongly decreased salivary gland invasion capacity (*aX-I* and *aL-I*) no SGS but 25,000 HLS were injected.

| Parasite line | Route of inoculation | Infected Mice[‡] | Prepatency (days) |
|---|---|---|---|
| *wild type ANKA* | by mosquito bite[§] | 4/4 | 3.0 |
| *wild type ANKA* | 10,000 HLS | 4/4 | 3.0 |
| *TRAP-I fluo* | by mosquito bite[§] | 4/4 | 3.0 |
| *TRAP-I fluo* | 500,000 MGS | 1/1[†] | 8.0 |
| *TRAP-I fluo* | 10,000 HLS (i.v.) | 4/4 | 4.0 |
| *TRAP-I fluo* | 10,000 SGS (i.v.) | 4/4 | 3.0 |
| *MIC2-I fluo* | by mosquito bite[§] | 4/4 | 3.0 |
| *MIC2-I fluo* | 10,000 HLS (i.v.) | 4/4 | 3.8 |
| *MIC2-I fluo* | 10,000 SGS (i.v.) | 8/8 | 3.1 |
| *αX-I fluo* | by mosquito bite[§] | 0/4 | /* |
| *αX-I fluo* | 500,000 MGS (i.v.) | 2/4 | 8.0 |
| *αX-I fluo* | 10,000 HLS (i.v.) | 1/4 | 6.0 |
| *αX-I fluo* | 25,000 HLS (i.v.) | 5/8 | 5.5 |
| *αL-I fluo* | by mosquito bite[§] | 0/4 | /* |
| *αL-I fluo* | 500,000 MGS (i.v.) | 0/2 | /* |
| *αL-I fluo* | 10,000 HLS (i.v.) | 0/4 | /* |
| *αL-I fluo* | 25,000 HLS (i.v.) | 1/6 | 5.0 |
| *trap(-)* | 500,000 MGS (i.v.) | 0/4 | /* |
| *trap(-)* | 25,000 HLS (i.v.) | 0/6 | /* |
| *trapΔI* | by mosquito bite[¶] | 0/4 | /* |
| *trapΔI* | 500,000 MGS (i.v.) | 0/6 | /* |
| *trapΔI* | 10,000 HLS (i.v.) | 0/4 | /* |
| *trapΔI* | 25,000 HLS (i.v.) | 0/4 | /* |
| *trapΔI* | 5,000 SGS (i.v.) | 0/4 | /* |
| *TRAP-I* | 10,000 HLS (i.v.) | 4/4 | 3.0 |
| *MIC2-I* | 10,000 HLS (i.v.) | 4/4 | 4.0 |
| *αX-I* | 10,000 HLS (i.v.) | 3/4 | 5.3 |
| *αL-I* | 10,000 HLS (i.v.) | 0/4 | /* |
| *RevCharge* | by mosquito bite[¶] | 2/8 | 5.5 |
| *RevCharge* | 10,000 SGS (i.v.) | 9/9 | 3.9 |

*mice did not become positive within ≥10 days post infection.

[†] three mice had to be sacrificed due to tail infections that occurred after injection.

[‡] Infected mice/inoculated (exposed) mice.

[§] mosquitoes were pre-selected for parasites.

[¶] mosquitoes were not pre-selected for parasites.

**Table 3.** Clustered summary of in vivo infections.

Comparison of the sporozoite infectivity to mice of the different lines by adding the data from *Table 2* where the respective controls were 100% infective. Data for wild type controls, *MIC2-I* domain and *RevCharge* parasites are from infections with hemolymph and salivary glands sporozoites as well as by bite experiments. All others summarize infections with hemolymph sporozoites only, as these parasites did not colonize the salivary glands efficiently.

| Parasite lines | Infected mice[*] |
|---|---|
| *wild type controls* | 24/24 |
| *MIC2-I domain* | 20/20 |
| *RevCharge* | 11/17 |
| *αX-I domain* | 9/16 |
| *αL-I domain* | 1/14 |
| *trapΔI* | 0/8[†] |
| *trap(-)* | 0/6 |

[*] Infected mice/inoculated (exposed) mice.

[†] Four additional mice injected with 5000 salivary gland sporozoites also remained uninfected.

for *αX-I* ranged from ~100 (*fluo*) to ~200 (*non-fluo*) and for *αL-I* from ~30 (*non-fluo*) to ~90 (*fluo*) sporozoites per gland (*Figure 3F,G* and *Table 1*) compared to 0–200 for *trap(-)* sporozoites and 0–800 for *trapΔI*. These small numbers might also be due to contaminants from the hemolymph and hence need to be treated with caution. Yet in all (8/8) *αX-I* mosquito infections sporozoites were detected in salivary glands, whereas this was only the case in 50% (4/8) of *αL-I* infections. In contrast for mosquitoes infected with *trap(-)* or *trapΔI* sporozoites were only detected in 13% (1/8) and 21% (3/14) experiments, respectively. (*Table 1*). In line with this observation, *αX-I* but not *αL-I* sporozoites could be detected in isolated salivary glands by live fluorescence microscopy (*Figure 3H* and *Video 1*) that indeed sporozoites expressing *αX-I* can enter this organ more efficiently than those expressing *αL-I*.

## Divergent I domains can partially restore gliding motility and infectivity of sporozoites

We next analyzed motility of the parasite lines in vitro using the classification scheme shown in *Figure 2—figure supplement 2*. Gliding assays of hemolymph sporozoites revealed that ~24% of *TRAP-I* but only ~4% of *MIC2-I* sporozoites were productively motile (*Figure 4A*). As expected, a higher proportion of salivary gland sporozoites were productively motile;~53% for *TRAP-I* and ~15% for *MIC2-I* sporozoites (*Figure 4B*). Among hemolymph sporozoites,~1% of *αX-I* (*Video 2*) but none of the >3000 observed *αL-I* sporozoites showed productive movement (*Figure 4A*). Motile *TRAP-I* and *MIC2-I* salivary gland sporozoites moved with a similar speed of ~1.5 µm/s (*Figure 4C*), showed similar trajectories (*Figure 4D*), and showed similarly persistent gliding (*Figure 4E*). Owing to the low numbers of *αX-I* and *αL-I* sporozoites in the glands (*Figure 3*), similar quantitation of sporozoite motility was not possible.

To probe the infectivity of mutant sporozoites, mice were exposed to infected mosquitoes. This revealed infection rates of 100% with a similar prepatent period (time until an infection could be detected in the blood) for *TRAP-I* and *MIC2-I* sporozoites (*Table 2*, *Table 3*). In contrast, no transmission could be observed for *αX-I* and *αL-I* sporozoites (*Figure 3—figure supplement 2*, *Table 2*). Intravenous injection of 10,000 *TRAP-I* or *MIC2-I* salivary gland sporozoites also infected all mice with a prepatency of three days. Unfortunately, the numbers of *αX-I* and *αL-I* salivary gland sporozoites were too low for comparative tests (*Table 1*). We therefore injected 10,000 hemolymph sporozoites, which again resulted in similar infection rates as seen for *TRAP-I* and *MIC2-I* sporozoites with a slightly longer prepatency when compared to salivary gland sporozoites (*Figure 3—figure supplement 3*, *Table 2*). Interestingly, 50% (4/8) of mice injected with 10,000 *αX-I* hemolymph sporozoites became blood stage patent after a delayed prepatency of >5 days, while no infection was observed for the same number of *αL-I* hemolymph sporozoites (0/8) (*Figure 3—figure supplement 3*, *Table 2*). When injecting 25,000 hemolymph sporozoites, 5 of 8 mice injected with *αX-I* sporozoites became

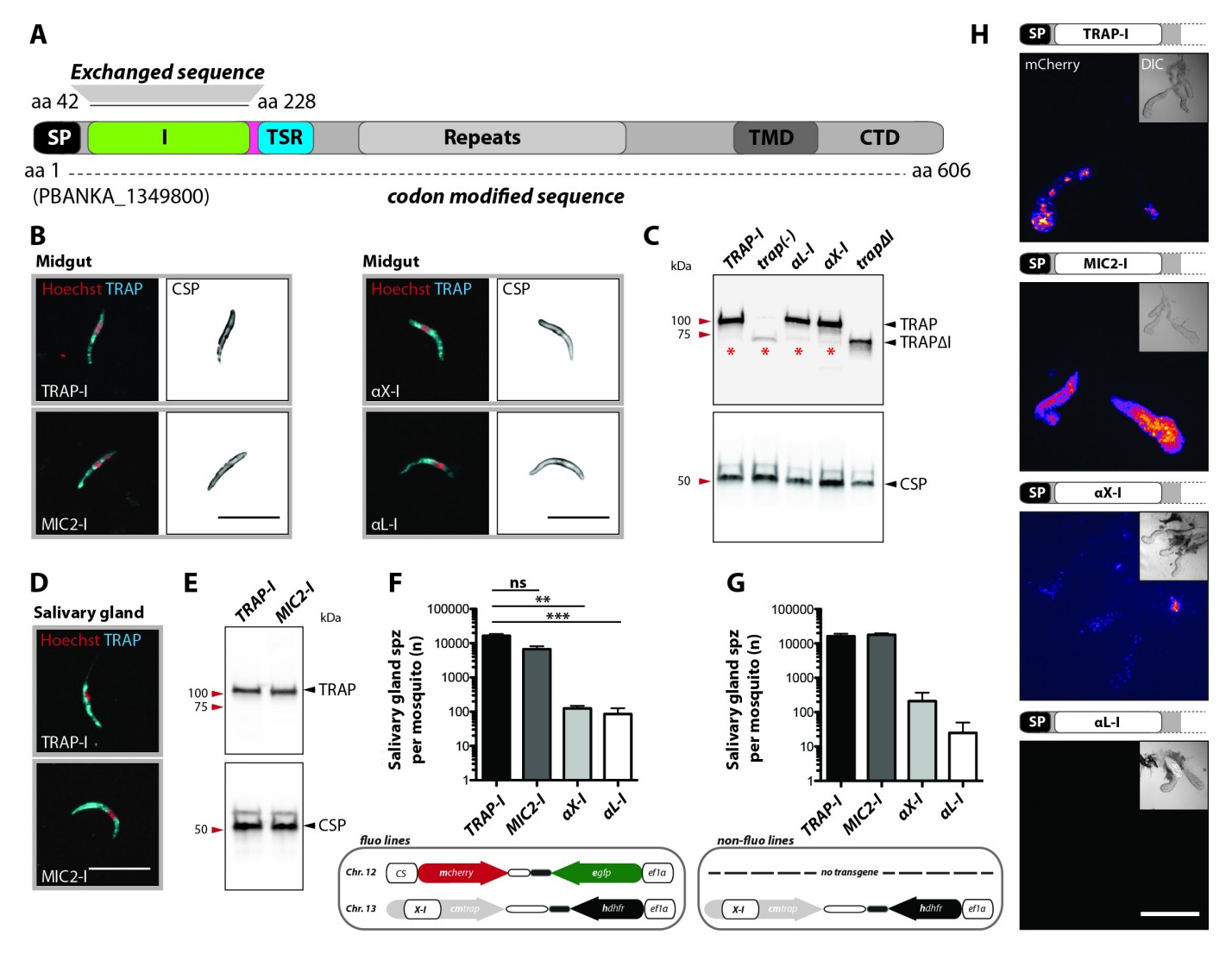

**Figure 3.** Sporozoites expressing TRAP with I domains from *T. gondii* or humans invade salivary glands at different levels. (A) Domain architecture of full-length TRAP (see *Figure 2A* legend) indicating the exchanged I domain. (B) Immunofluorescence assay (IFA) using antibodies against TRAP and CSP on non-fluorescent permeabilized *TRAP-I*, *MIC2-I*, *aX-I* and *aL-I* midgut sporozoites. Scale bars: 10 μm. (C) Western blot probing TRAP from control (*TRAP-I*), *trap(-)*, *trapΔI*, *aX-I* and *aL-I* hemolymph sporozoites. Anti-CSP antibodies were used as loading control. The asterisks mark an unspecific band observed in lysates of hemolymph sporozoites but not in lysates of salivary gland sporozoites (see panel E for TRAP-I). Repeated PCR of genomic DNA from the parasite lines also suggested that this band is not due to a TRAP fragment. (D) IFA using anti-TRAP antibodies on non-fluorescent *TRAP-I* and *MIC2-I* salivary gland sporozoites. Scale bars: 10 μm. (E) Western blot probing TRAP and CSP of *TRAP-I* and *MIC2-I* salivary gland sporozoites. (F, G) Quantification of salivary gland sporozoites of fluorescent (*fluo*) (F) and non-fluorescent (*non-fluo*) (G) I domain mutants 17–24 days post infection. The pictogram below each graph indicates the genotype of the tested parasite lines. Shown is the mean ± SEM of at least three (F) or two (G) countings from three (F) and one (G) independent feeding experiments. **p<0.05 one-way-ANOVA (Kruskal-Wallis-test). Note that mosquitoes infected with *non-fluo* lines were not preselected for fluorescent parasites before dissection potentially yielding smaller numbers. No significance test for (G) as data derived from one experiment. (H) Salivary gland colonization by sporozoites expressing the different I domains shown above each image. The fluorescence of mCherry expressing sporozoites is coded for intensity. The small image inset depicts the respective salivary gland in differential interference contrast (DIC). Scale bar: 200 μm. Invasion of the salivary glands by *aX-I* sporozoites is also shown in *Video 1*.

The online version of this article includes the following figure supplement(s) for figure 3:

**Figure supplement 1.** Generation of *P. berghei* strains expressing TRAP with different I domains.

**Figure supplement 2.** Transmission efficacy of salivary gland sporozoites expressing TRAP with different I domains transmitted either by mosquito bite or by intravenous injection.

**Figure supplement 3.** Transmission efficacy of intravenously injected hemolymph sporozoites expressing TRAP with different I domains.

**Figure supplement 4.** Sporozoites expressing the integrin I domains αX or αL are infective to mice.

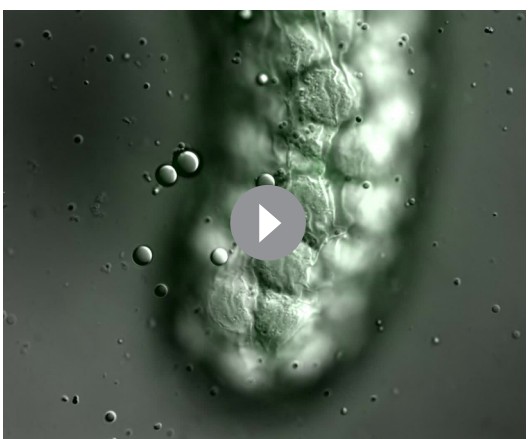

**Video 1.** Z-projection through salivary gland infected with *aX-I fluo* sporozoites. Shown is an image series in Z-direction of a salivary gland infected with *aX-I fluo* sporozoites. Images were taken on an Axiovert 200M (Zeiss) with a 63x (N.A. 1.3) objective.
https://elifesciences.org/articles/57572#video1

blood stage patent and additionally, 1 of 6 mice injected with $\alpha$L-I sporozoites became infected (*Figure 3—figure supplement 3*, *Table 2*, *Table 3*). Finally, we injected mice with 500,000 midgut sporozoites to compare $\alpha$X-I and $\alpha$L-I side by side with *trap(-)* and *trap$\Delta$I*. Injections of $\alpha$X-I midgut sporozoites infected 2 out of 4 mice with a prepatent period of 8 days. In contrast no infections in mice could be observed when $\alpha$L-I (0/2), *trap(-)* (0/4) or *trap$\Delta$I* (0/6) midgut sporozoites were injected (*Table 2*). Mice infected with the same number of wild type or wild type-like midgut sporozoites become blood stage patent after 6–8 days (*Table 2*; *Klug and Frischknecht, 2017*). To exclude spurious results from contamination with other parasite lines, some of the transmitted parasites were propagated in mice and analyzed via PCR for the correct genotype (*Figure 3—figure supplement 4*). Additionally, the TRAP locus of these parasites was sequenced to ensure that the correct I domain was present. In all tested cases, the expected $\alpha$X-I and $\alpha$L-I genotype was confirmed. Thus, *TRAP-I* and *MIC2-I* sporozoites are comparably infectious to mice, $\alpha$X-I sporozoites are moderately infectious and $\alpha$L-I, while nearly completely deficient in infectivity, could in one case still cause an infection.

## Sporozoites expressing the MIC2 I domain are impaired in hepatocyte invasion but do not show altered host cell tropism

While *Plasmodium* sporozoites can infect different types of cells they have a strong tropism for the liver. In contrast, *Toxoplasma gondii* can infect any nucleated cell from a warm-blooded animal (*Boothroyd, 2009*). Hence, we tested whether an exchange of the I domain affected host cell invasion or tissue tropism of sporozoites. First, we tested the capacity of *TRAP-I* and *MIC2-I* salivary gland sporozoites to invade HepG2 cells. After exposure for 1.5 hr, 2-fold fewer *MIC2-I* sporozoites than *TRAP-I* sporozoites were intracellular (*Figure 5A*). At 48 hr after infection, 5-fold fewer *MIC2-I* sporozoites had developed into liver stage parasites when compared to *TRAP-I* sporozoites (*Figure 5B*). However, the size of liver stage parasites after 48 hr was comparable implying that an exchange of the I domain affects parasite invasion but not intracellular development (*Figure 5C,D*).

To test in vivo tropism of sporozoites, mice were infected by intravenous injection of 20,000 *TRAP-I* and *MIC2-I* salivary gland sporozoites and after 42 hr liver, spleen, lung and a part of the small intestine were harvested. Parasite load was measured by quantitative RT-PCR. Liver tropism of both *TRAP-I* and *MIC2-I* salivary gland sporozoites was pronounced, with ~16 fold more parasites localizing to the liver than to any other organ (*Figure 5E*, *Figure 5—figure supplement 1*). However, the liver burden of *MIC2-I* parasites was reduced by ~40% relative to *TRAP-I* parasites, reflecting the similar decrease observed during in vitro infection experiments.

## Charge reversal mutations of the TRAP I domain differentially impact infectivity

Based on the results with different I domains, we sought a common pattern explaining the observed phenotypes. One parameter important for protein-protein interactions is surface charge. Interestingly, the surface of the TRAP I domain is very basic, with a pI of 9.7 (*Figure 6A*), while the surface of the second best functioning I domain, MIC2, is acidic (pI 6.1) (*Figure 6B*). Of the human integrin I domains, the one from $\alpha$X has a basic charge (pI 8.9), similar to the TRAP I domain (*Figure 6C*), while the one of $\alpha$L is even slightly more negatively charged (pI 5.8) (*Figure 6D*) than the I domain of MIC2. To test if surface charge could be important for *P. berghei* TRAP I-domain function, we rendered it anionic (pI of 6.8) with seven charge reversal mutations (H56E, H62E, H123E, K164Q,

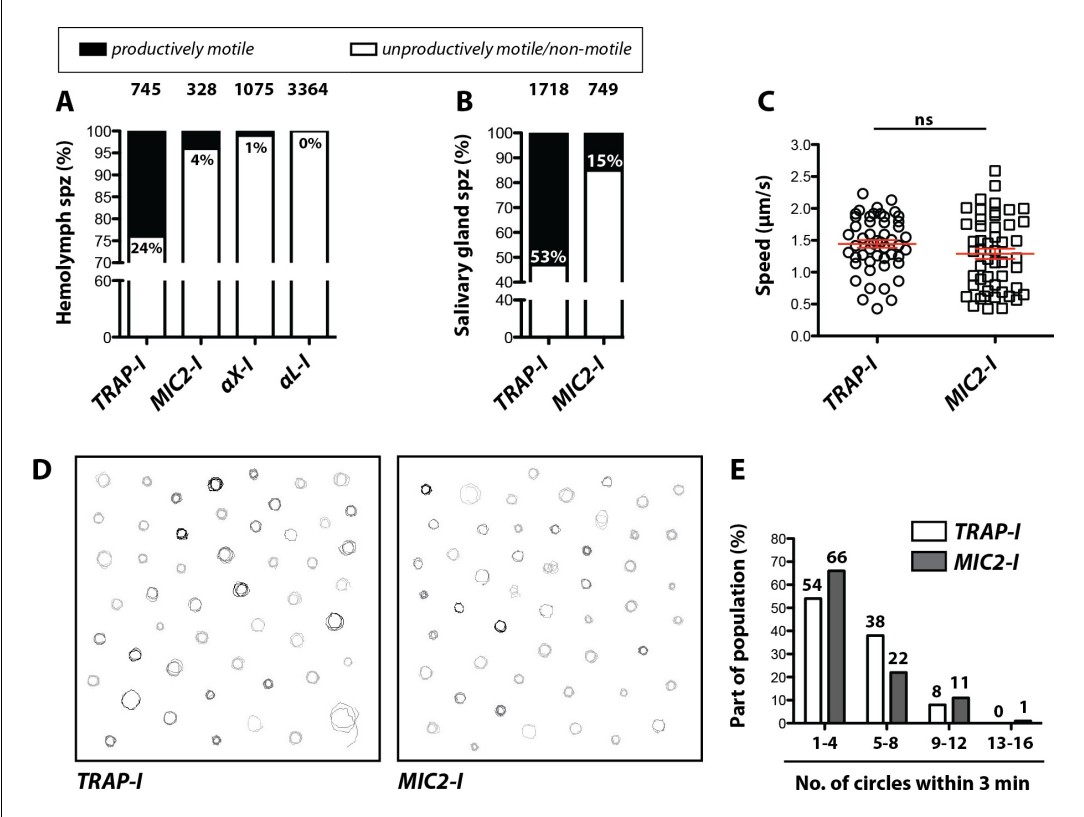

**Figure 4.** Impaired in vitro gliding of I domain mutant sporozoites. (A, B) Ratio of productively and unproductively moving/non motile hemolymph (A) and salivary gland (B) sporozoites of the indicated parasite lines. Sporozoites were classified as productively moving if they were able to glide at least one complete circle during a three-minute movie. All sporozoites that behaved differently were classified as unproductively moving/non motile. Data were generated from three independent experiments per parasite line. The number of analyzed sporozoites is depicted above each column. For further details about movement pattern of sporozoites see *Figure 2—figure supplement 2*. Note that salivary gland sporozoites from the lines αX-I and αL-I could not be analyzed because of their low number. (C) Average speed of salivary gland sporozoites continuously gliding for 150 s. Shown are the mean speeds of 50 salivary gland sporozoites per line generated from two (*MIC2-I*) and three (*TRAP-I*) independent experiments. Red bars show the mean ± SEM; unpaired t-test. (D) Trajectories of the sporozoites tracked in (C). (E) Persistence of gliding *TRAP-I* and *MIC2-I* sporozoites. The graph illustrates the number of circles salivary gland sporozoites were able to glide during a three-minute movie. Data were generated from two (*MIC2-I*; 107 sporozoites) and three (*TRAP-I*; 111 sporozoites) independent experiments.

K165D, R195E and K202E) around the perimeter of the putative ligand binding site, distal from the MIDAS (*Figure 6E*, *Figure 6—figure supplement 1*). Secretion of TRAP to the surface was not altered in these *RevCharge* sporozoites and no difference in protein sub-cellular distribution or expression was observed (*Figure 6F,G*). Strikingly, salivary gland invasion was also not affected (*Figure 6H*, *Table 1*). However, only ~1% of *RevCharge* salivary gland derived sporozoites were productively motile in gliding assays compared to ~53% of *TRAP-I* parasites (*Figure 6I*). Infection by mosquito bite revealed that only 2 out of 8 mice in two independent experiments became blood stage positive with a delay in prepatency of >2 days (*Figure 6J*, *Table 2*, *Table 3*) indicating a decreased infectivity of salivary gland sporozoites by over 99%. In

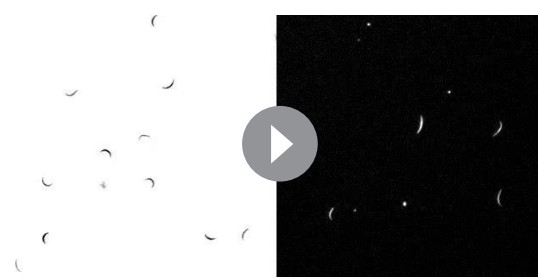

**Video 2.** *TRAP-I* and αX-I hemolymph sporozoites gliding in vitro. Movie showing hemolymph sporozoites expressing mCherry of *TRAP-I* (shown on the left with a white background) and αX-I (shown on the right with a black background) productively gliding in vitro. Imaging was performed on an Axiovert 200M (Zeiss) with a 10x (NA 0.25) objective. Shown is a three-minute movie with three seconds between frames.
https://elifesciences.org/articles/57572#video2

contrast, injection of 10,000 salivary gland sporozoites in three independent experiments (nine mice in total) led to an infection of all mice with the prepatency period of the *RevCharge* mutant being delayed by 0.5–1 day compared to controls (wild type/*TRAP-I*) (*Figure 6K*, *Table 2*). This corresponds to a decreased infectivity of 50–90%.

In vitro invasion of liver cells also showed a severe impairment of the *RevCharge* mutant compared to *TRAP-I*. While after 2 hr 54% of *TRAP-I* sporozoites showed an intracellular localization only 3% of *RevCharge* sporozoites were observed inside cells (*Figure 6—figure supplement 2A*). In line with this result very few growing liver stages of the *RevCharge* mutant were observed while on average >250 liver stages per well were counted for *TRAP-I* (*Figure 6—figure supplement 2B*). However, liver stages of the *RevCharge* mutant developed normally (*Figure 6—figure supplement 2C*). These results suggest that the basic charge around the MIDAS of the TRAP I domain is a key determinant of sporozoite motility and infectivity during natural transmission from mosquito to mammal. Most intriguingly, the introduced mutations appear to uncouple the capacity of the *P. berghei* sporozoite to infect insect salivary glands from efficiently infecting the rodent host.

## Discussion

### Structurally related I domains can rescue TRAP function

Previous studies on the function of the TRAP I domain focused on invariant residues that coordinate the MIDAS metal ion in the I domain (*Wengelnik et al., 1999*; *Matuschewski et al., 2002*). The sidechains of these five invariant I domain residues coordinate the MIDAS $Mg^{2+}$ ion either directly or indirectly through water molecules. In contrast to the closed conformation, in the open conformation of TRAP and integrin αI domains, neither of the two Asp residues directly coordinate the MIDAS metal ion. The metal ion is therefore thought to have high propensity in the open conformation to bind an acidic residue in the ligand. The MIDAS residues and their bound water molecules occupy five of the six coordination positions around the MIDAS $Mg^{2+}$ ion. The remaining, sixth coordination position is occupied when a critical Asp or Glu sidechain in the ligand binds through a carboxyl oxygen to the MIDAS $Mg^{2+}$ ion (*Liddington, 2014*). Mutation of single MIDAS residues or removal of the $Mg^{2+}$ ion by chelation abolishes ligand binding by integrins (*Michishita et al., 1993*; *Kern et al., 1994*; *Kamata et al., 1995*). Similarly, mutation of the MIDAS motif of TRAP severely impairs salivary gland invasion and infectivity of the vertebrate host while gliding motility is decreased in salivary gland but not in hemolymph sporozoites (*Matuschewski et al., 2002*). In contrast, deletion of TRAP completely abrogates salivary gland invasion, infectivity and productive motility (*Sultan et al., 1997*; *Münter et al., 2009*) and severely affects substrate adhesion (*Münter et al., 2009*; *Hegge et al., 2010*).

To elucidate the role of the I domain in TRAP function, we generated the parasite line *trapΔI* expressing TRAP without its I domain. TRAPΔI was expressed and correctly localized in sporozoites as shown by western blotting and immunofluorescence. Interestingly, this line was severely deficient in salivary gland invasion and could not infect mice. Furthermore, we observed no productive movement and only some back-and-forth motility in *trapΔI* hemolymph sporozoites, similar to *trap(-)* hemolymph sporozoites. These results suggest that the biological functions of TRAP are dependent on its I domain. It was therefore remarkable that TRAP function was largely restored by replacement of its I domain by its homologue from *Toxoplasma gondi* with only 28% amino acid sequence identity. Compared to cell surface receptors that engage in typical protein-protein interactions, that is those that are not dependent on $Mg^{2+}$ ions, these sequence identities are substantially below the level of 40% to 50% identity generally required for members of the same receptor family to recognize the same ligand. The strength of the $Mg^{2+}$ ion bond to the oxygen in the ligand, which has a bond distance of only 2 Å and is partially covalent, may overcome the lack of complementarity in other regions of the ligand binding site, and explain the ability of I domains of such remarkably low sequence identity to function in TRAP. In addition to the low sequence identity, there are six to seven sequence positions where residues are inserted or deleted in Toxoplasma MIC2 or human integrin αI domains compared to TRAP (*Figure 1F*). Nonetheless, all regions involved in conformational change around the MIDAS and in the β6-α7 loop are highly conserved in conformation in both the open and closed conformations of the apicomplexan and human I domains (*Song et al., 2012*; *Song and Springer, 2014*). Western blotting showed that the αL-I, αX-I, and MIC2-I TRAP fusions

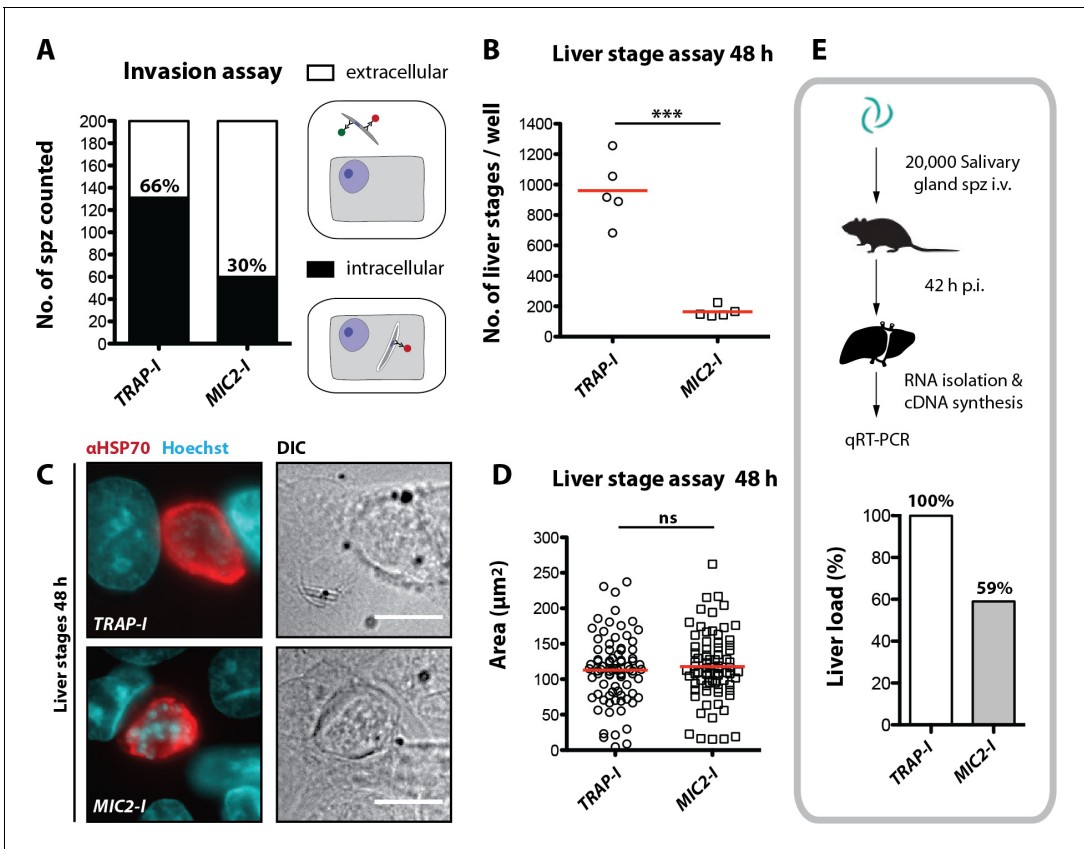

**Figure 5.** *MIC2-I* sporozoites are impaired in hepatocyte invasion but not in liver stage development. (**A**) Invasion assay. Confluent HepG2 monolayers exposed to sporozoites for 1.5 hr were fixed and stained with CSP antibodies before and after permeabilization with methanol; anti-IgG secondary antibodies conjugated to different fluorochromes were used before and after permeabilization. The illustration on the right depicts the staining of intracellular and extracellular sporozoites. (**B**) Number of liver stages after challenge with 10,000 salivary gland sporozoites. Experiments were performed with sporozoites obtained from two different feeding experiments with 2–3 technical replicates per parasite line and experiment. The red line indicates the median. *** depicts a p-value of 0.005; unpaired t test. (**C, D**) Liver stage development of *TRAP-I* and *MIC2-I* parasites. (**C**) Immunofluorescence images of *TRAP-I* and *MIC2-I* liver stages 48 hr post infection. The expression of *Pb*HSP70 is shown in red while DNA is shown in cyan. Scale bars: 10 µm. (**D**) Area of liver stages 48 hr post infection. The red line shows the median. Data were tested for significance with an unpaired t test. (**E**) Liver load in mice challenged with *MIC2-I* as % of load with the control *TRAP-I*. Shown is the relative expression of parasite-specific 18 s rRNA determined by qRT-PCR and normalized to mouse-specific GAPDH. Data display the mean of three measurements generated from pooled cDNA of four challenged mice per parasite strain.

The online version of this article includes the following figure supplement(s) for figure 5:

**Figure supplement 1.** *MIC2-I* sporozoites show no altered tissue tropism compared to *TRAP-I*.

were as well expressed as TRAP-I; furthermore, immunofluorescence showed normal localization in sporozoites, suggesting that trafficking was not impaired.

Although deletion of the TRAP I domain completely abolished mosquito salivary gland invasion and mouse infection by midgut and hemolymph sporozoites, the MIC2 I domain completely restored these functions. Moreover, no differences in mouse infection were observed between TRAP and MIC2 I domain replacements in infection by mosquito bite or intravenous injection with salivary gland sporozoites, including the length of the prepatent period. Fewer *MIC2-I* than *TRAP-I* sporozoites were able to glide productively in vitro; however, the gliding speed of motile sporozoites and persistence of gliding was similar. Infectivity of HepG2 liver cells was significantly decreased for *MIC2-I* compared to *TRAP-I* sporozoites; yet, quantitative RT-PCR showed that there was little difference in ability to infect liver cells in vivo between intravenously injected MIC2-I and TRAP-I sporozoites. Moreover, there was no change in in vivo tropism.

We also examined replacement with two mammalian integrin I domains, from the promiscuous αXβ2 integrin and the much more selective αLβ2 integrin. Salivary gland invasion was decreased 100

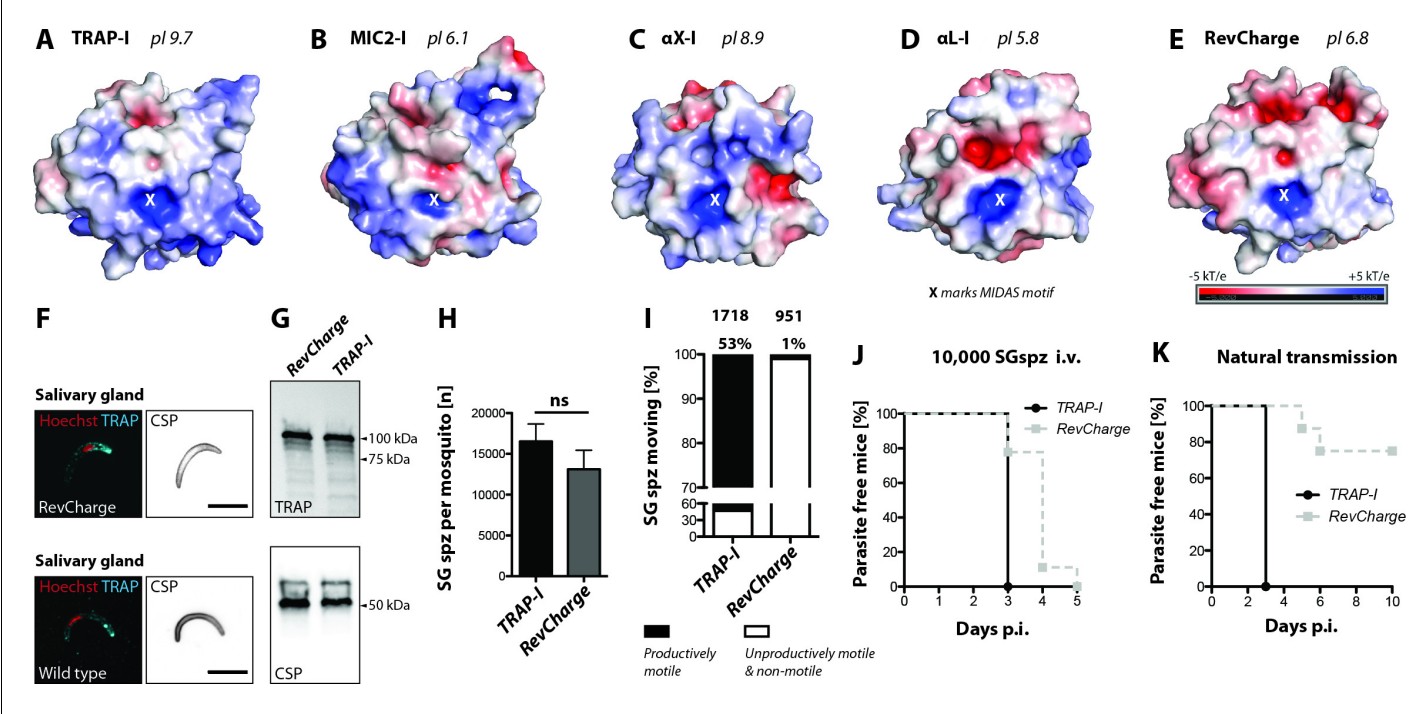

**Figure 6.** Sporozoites expressing an I domain with a negatively charged MIDAS perimeter invade salivary glands normally but show decreased motility and infectivity to mice. (A–E) Electrostatic surfaces around the MIDAS metal ion of I domains in the open conformation. Structures are of (A), *P. berghei* TRAP modeled on *P. vivax*, PDB 4HQL; (B), MIC2, modeled on closed MIC2 (PDB 4OKR chain B) and open TRAP (PDB 4HQL); (C), integrin αX, PDB 4NEH; (D), integrin αL, PDB 1MQ8; and (E), *P. berghei* RevCharge modeled on *P. vivax*, (PDB 4HQL). The X marks the MIDAS. The color code refers to the electrostatic potential of the surface ranging from −5 to +5 kT/e. F) IFA on unpermeabilized salivary gland sporozoites of *RevCharge* and wild type (*wt*); cyan: TRAP, red: DNA, CSP: black. Scale bars: 10 μm. (G) anti-TRAP western blot of salivary gland control (*TRAP-I*) and *RevCharge* sporozoites; loading control: CSP. (H) Sporozoite numbers in salivary glands of mosquitoes infected with *TRAP-I* (*fluo*) and *RevCharge* (*non-fluo*) from three independent feeding experiments; Mann-Whitney test. Note that data for the control *TRAP-I* (*fluo*) are shown in **Figure 3F**. (I) Movement pattern of *TRAP-I* and *RevCharge* salivary gland sporozoites from three different feeding experiments. Sporozoites were classified as productively moving if they were able to perform ≥1 circle during a three minutes movie. All sporozoites that behaved differently were classified as unproductively moving or non-motile. Note that data for the control TRAP-I were shown previously in **Figure 4B**. (J, K) In vivo infectivity of *TRAP-I* and *RevCharge* salivary gland sporozoites. C57BL/6 mice were either exposed to infected mosquitoes or injected with 10,000 salivary gland sporozoites/mouse intravenously (i.v.) (see also **Table 2**). (J) The percentage of parasite-free mice over time after injection of 10,000 salivary gland sporozoites (TRAP-I, n = 4; RevCharge, n = 9). (K) The percentage of parasite-free mice over time after exposure to infected mosquitoes (TRAP-I, n = 4; RevCharge, n = 8). p.i. – post infection.

The online version of this article includes the following figure supplement(s) for figure 6:

**Figure supplement 1.** Alignment of TRAP I domain homologues of different *Plasmodium* species in comparison to the I domain of the *PbRevCharge* mutant.

**Figure supplement 2.** *RevCharge* sporozoites show impaired hepatocyte invasion but normal liver stage development.

to 400 and 500 to 1,000-fold in αX-I and αL-I compared to *TRAP-I* sporozoites, respectively. Infectivity of hemolymph αX-I sporozoites in mice was 100-fold deacreased compared to controls while αL-I hemolymph sporozoites showed nearly no infectivity as *trap(-)* sporozoites. The greater efficacy of αX-I than αL-I sporozoites in invasion of mosquito salivary glands, infection of mice, and productive motility in vivo confirmed our hypothesis that the polyreactive αX I domain would better support TRAP function than the highly specific αL I domain, albeit at low levels.

Multiple factors may account for the substantially lesser efficacy of mammalian integrin I domains than the MIC2 I domain. MIC2 has a similar function to TRAP in *Toxoplasma* and may be its orthologue. Each has an I domain that is inserted in an extensible β-ribbon in tandem with a TSR domain and a long-range disulfide bond that moves in allostery (**Figure 1**). Integrins, TRAP, and MIC2 connect to the actin cytoskeleton, which applies force to their cytoplasmic domains that is resisted by ligands and provides traction for motility and cell invasion. The force (F) transmitted through the I domain, times the difference in extension of the I domain in the closed and open states (Δx), gives

an energy (F•Δx = E) that tilts the energy landscape toward the open, high affinity state (*Li and Springer, 2017*). Because of their extensible β-ribbons, Δx is substantially greater for TRAP and MIC2 than for integrin I domains, and cytoskeletal forces may also differ considerably. To compensate for these differences, the energy differences between the closed and open states, as well as the kinetics for crossing between them, may be tuned differently for TRAP and MIC2 than for integrin I domains. Their degree of polyspecificity is also likely to vary. Integrins αXβ2 and αMβ2 have each been reported to bind >30 ligands, including glycoproteins and heparan (*Yakubenko et al., 2002*). However, these integrin I domains can also bind specifically, as shown by binding to distinct high-affinity sites within their common ligand, iC3b (*Xu et al., 2017*).

The most polyreactive integrin I domains, αX and αM are basic with higher pI (8.9 and 9.5, respectively) than the highly specific αL (pI 5.6), αE (pI 4.6), and collagen-binding α1, α2, α10, and α11 I domains (pI 5.1–5.6). As the surfaces of all cells are negatively charged and almost all proteins are acidic, we tested whether the functions of the basic TRAP I domain (pI = 9.7) would be altered with seven substitutions that made it neutral (pI 6.8). This *RevCharge* mutant was well expressed and localized. *RevCharge* sporozoites accumulated well in mosquito salivary glands and were infectious when injected intravenously. However, gliding in vitro and infectivity by mosquito bite were greatly decreased. Although the pI of RevCharge of 6.8 was close to that of MIC2, its mutations were clustered around the periphery of the MIDAS site and the electrostatic surface of its ligand-binding face more closely resembled that of αL than MIC2, TRAP, or αX (*Figure 6*). These results suggest that the distribution of electrostatic charge on the TRAP I domain is an important variable. The difference between infection by mosquito bite and intravenous injection further suggests that electrostatics may have an important role in successful sporozoite exit from the dermis into the circulation.

The findings that the I domain from *Toxoplasma* MIC2 that encounters different ligands as TRAP can replace its function in mediating salivary gland invasion and infection of mice strongly suggest that the I domain functions to provide traction for motility and cell invasion rather than tissue tropism. Binding of the TRAP I domain to multiple distinctive ligands is required by its function in mosquitos in migration into salivary glands and in vertebrate hosts in emigration from the dermis into the circulation and from the circulation into the liver, followed by productive invasion of liver parenchymal cells. The ligands in the mosquito must differ from those in the vertebrate; those encountered in the vertebrate in the dermis, endothelium and liver may also differ. Furthermore, *Toxoplasma* lives in epithelia in the gastrointestinal system of cats and other mammals, and lacks an arthropod host, yet its MIC2 I domain functions robustly in mosquito salivary gland invasion. Our results are consistent with TRAP binding to the many ligands that have been identified for it, including liver-specific fetuin-A (*Jethwaney et al., 2005*), proteoglycans (*Robson et al., 1995*; *Pradel et al., 2002*), glycosaminoglycans (*Matuschewski et al., 2002*), and the integrin αV subunit (*Dundas et al., 2018*) in vertebrates and saglin in mosquitos (*Ghosh et al., 2009*).

Abolition of TRAP function by deletion of its I domain as shown here, together with the negligible effects of mutation or exchange of other domains (*Matuschewski et al., 2002*; *Ejigiri et al., 2012*) strongly suggests that the I domain is the sole ligand binding domain. A model in which TRAP does not provide liver tropism is also consistent with the expression of TRAP in sporozoites of *Plasmodium* species that infect birds and initially infect skin rather than liver cells (*Böhme et al., 2018*).

Our results support a model in which TRAP does not determine tropism for salivary glands in mosquitos or liver in mammals and instead acts as a poly-specific receptor that provides traction for sporozoite emigration into these tissues and infection of cells that is triggered by other receptors. Similarly, integrins, from which the TRAP I domain may have been borrowed, are activated by cytoskeletal activity, which is generally signaled by G protein-coupled receptors or receptors that couple to tyrosine kinases. This model has the advantage that the signaling receptors can be extremely sensitive and ATP-dependent cytoskeleton polymerization or actomyosin contraction through activation of adhesins can greatly amplify those signals and provide ultrasensitive regulation of adhesiveness (*Li and Springer, 2017*). This might be essential during the stick-and-slip gliding of sporozoites (*Münter et al., 2009*).

Chemoattractants have the advantage that they can drive directional migration through multiple layers of distinctive cell types as in the liver. In vertebrates, chemoattractants and their receptors cooperate with integrins and their ligands to enable highly specific leukocyte emigration from the vasculature into organs or sites of inflammation (*Springer, 1994*). Chemoattractant receptors evolved distinctly in prokaryotes and eukaryotes and perhaps take yet a different form in

apicomplexans. The ability of chemoattractants to activate motility, termed chemokinesis, may be related to the use here of BSA, a known carrier of fatty acids and other hydrophobes, to activate sporozoite gliding. Our finding that the I domain of TRAP is required for gliding motility and invasion by sporozoites of mosquito salivary glands and mammalian liver, without a strong requirement for evolved specificity, suggests that new paradigms may be needed to understand sporozoite activation and homing of sporozoites to specific organs in their hosts.

# Materials and methods

## Key resources table

| Reagent type (species) or resource | Designation | Source or reference | Identifiers | Additional information |
|---|---|---|---|---|
| Antibody | anti-CSP (mouse monoclonal) | *Yoshida et al., 1980* | MR4: MRA-100 | mAb 3D11 |
| Antibody | anti-TRAP (rabbit ployclonal) | This paper | / | / |
| Antibody | anti-rabbit (Goat) | ThermoFisher Scientific | Cat#A-11034 | coupled to AlexaFluor 488 |
| Antibody | anti-rabbit (Goat) | ThermoFisher Scientific | Cat#A10523 | coupled to Cy5 |
| Antibody | anti-mouse (Goat) | ThermoFisher Scientific | Cat#A-11001 | coupled to AlexaFluor 488 |
| Antibody | anti-mouse (Goat) | ThermoFisher Scientific | Cat#A10524 | coupled to Cy5 |
| Antibody | anti-rabbit (Goat) | Bio-Rad | Cat#1705046 | Immun Star (GAR)-HRP |
| Antibody | anti-mouse (sheep) | GE-Healthcare | NXA931-1ML | IgG, HRP linked whole Ab |
| Bacteria (*Escherichia coli*) | XL1-blue cells | Agilent technologies | Cat#200236 | Chemically competent cells |
| Other | Hoechst 33342 | ThermoFisher Scientific | Cat#H3570 | / |
| Commercial assay or kit | SYBR Green PCR Master Mix | ThermoFisher Scientific | Cat#4309155 | / |
| Cell line (*Homo sapiens*) | HepG2 | ATCC | HB-8065 | / |
| Strain (*Mus musculus*) | NMRI | Janvier Labs/Charles River Laboratories | / | / |
| Strain (*Mus musculus*) | C57BL/6JRj | Janvier Labs/Charles River Laboratories | / | / |
| Strain (*Plasmodium berghei*) | ANKA | *Vincke and Bafort, 1968* | MR4: MRA-671 | / |
| Strain (*Plasmodium berghei*) | *trap(-)rec* | this paper | / | ANKA background |
| Strain (*Plasmodium berghei*) | *trapΔI* | this paper | / | ANKA background |
| Strain (*Plasmodium berghei*) | *fluo* | this paper | / | ANKA background |
| Strain (*Plasmodium berghei*) | *TRAP-I fluo* | this paper | / | ANKA background |
| Strain (*Plasmodium berghei*) | *MIC2-I fluo* | this paper | / | ANKA background |
| Strain (*Plasmodium berghei*) | *αX-I fluo* | this paper | / | ANKA background |
| Strain (*Plasmodium berghei*) | *αL-I fluo* | this paper | / | ANKA background |

*Continued on next page*

*Continued*

| Reagent type (species) or resource | Designation | Source or reference | Identifiers | Additional information |
|---|---|---|---|---|
| Strain (*Plasmodium berghei*) | *TRAP-I non-fluo* | this paper | / | ANKA background |
| Strain (*Plasmodium berghei*) | *MIC2-I non-fluo* | this paper | / | ANKA background |
| Strain (*Plasmodium berghei*) | *αX-I non-fluo* | this paper | / | ANKA background |
| Strain (*Plasmodium berghei*) | *αL-I non-fluo* | this paper | / | ANKA background |
| Strain (*Plasmodium berghei*) | *RevCharge non-fluo* | this paper | / | ANKA background |
| Sequenced-based reagent | *gapdh* forward | this paper | PCR primers | TGAGGCCGGTGCTGAGTATGTCG |
| Sequenced-based reagent | *gapdh* reverse | this paper | PCR primers | CCACAGTCTTCTGGGTGGCAGTG |
| Sequenced-based reagent | 18 s RNA forward | this paper | PCR primers | AAGCATTAAATAAAGCGAATACATCCTTAC |
| Sequenced-based reagent | 18 s RNA reverse | this paper | PCR primers | GGAGATTGGTTTTGACGTTTATGTG |
| Recombinant DNA reagent | TRAP gene sequence: *TRAP-I* | ThermoFisher Scientific | / | codon modified (*E. coli* K12) |
| Recombinant DNA reagent | TRAP gene sequence: *MIC2-I* | ThermoFisher Scientific | / | codon modified (*E. coli* K12) |
| Recombinant DNA reagent | TRAP gene sequence: *αX-I* | ThermoFisher Scientific | / | codon modified (*E. coli* K12) |
| Recombinant DNA reagent | TRAP gene sequence: *αL-I* | ThermoFisher Scientific | / | codon modified (*E. coli* K12) |
| Recombinant DNA reagent | TRAP gene sequence: *RevCharge* | ThermoFisher Scientific | / | codon modified (*E. coli* K12) |
| Recombinant DNA reagent | pMK-RV | ThermoFisher Scientific | / | Kan[R] |
| Recombinant DNA reagent | Pb238 | *Deligianni et al., 2011* *Singer et al., 2015* | / | Amp[R] |
| Recombinant DNA reagent | PbGEM-107890 | *Schwach et al., 2015* PlasmoGEM | / | https://plasmogem.sanger.ac.uk/designs/search_result?id=PbGEM-107890 |
| Software, algorithm | Prism 5.0 | GraphPad, San Diego | / | https://www.graphpad.com/scientific-software/prism/ |
| Software, algorithm | PyMOL | The PyMOL Molecular Graphics System, Version 2.0 Schrödinger, LLC | / | https://pymol.org/2/ |
| Software, algorithm | AxioVision | Carl Zeiss Microscopy | / | https://www.zeiss.com/microscopy/int/home.html |
| Software, algorithm | Volocity | PerkinElmer | / | http://www.perkinelmer.de/corporate |
| Software, algorithm | ApE | ApE – A plasmid Editor by M. Wayne Davis | / | http://jorgensen.biology.utah.edu/wayned/ape/ |
| Software, algorithm | ImageJ | *Schindelin et al., 2012* | / | https://imagej.nih.gov/ij/ |
| Software, algorithm | Clustal Omega | *Sievers et al., 2011* | / | https://www.ebi.ac.uk/Tools/msa/clustalo/ |
| Software, algorithm | Optimizer | *Puigbò et al., 2007* | / | http://genomes.urv.es/OPTIMIZER/ |

*Continued on next page*

*Continued*

| Reagent type (species) or resource | Designation | Source or reference | Identifiers | Additional information |
|---|---|---|---|---|
| Software, algorithm | PlasmoDB (version 26–34) | *Aurrecoechea et al., 2009* | / | http://plasmodb.org/plasmo/ |

## Bioinformatic analysis

*Plasmodium* sequences were retrieved from PlasmoDB (http://plasmodb.org/plasmo/, version 26–34) (*Aurrecoechea et al., 2009*) and multiple sequence alignments were performed with Clustal Omega (http://www.ebi.ac.uk/Tools/msa/clustalo/) (*Sievers et al., 2011*). To change the codon usage of open reading frames (ORFs) we applied the tool OPTIMIZER (http://genomes.urv.es/OPTI-MIZER/) (*Puigbò et al., 2007*).

## Generation of *trap(-)* and *trapΔI* parasites

TRAP knockout (*trap(-)*) parasites were generated with the PlasmoGem (*Schwach et al., 2015*) vector (PbGEM-107890) using standard protocols (*Janse et al., 2006*). Isogenic *trap(-)* parasites were subsequently negatively selected with 5-fluorocytosine (1.0 mg/mL in the drinking water) to give rise to selection marker free *trap(-)* parasites (*Braks et al., 2006*). For the generation of *trapΔI* parasites we made use of the Pb238 vector (*Deligianni et al., 2011*; *Klug and Frischknecht, 2017*). In a first step the *trap* 3'UTR (970 bp) was amplified with the primers P165/P166 and cloned (*BamHI* and *EcoRV*) downstream of the resistance cassette in the Pb238 vector. In a next step the coding sequence of the *trap* gene including the 5' and 3' UTR was amplified with the primers P508/P509 and cloned in the pGEM-T-Easy vector giving rise to the plasmid pGEM-TRAPfull. Subsequently the pGEM-TRAPfull plasmid was mutated with the primers P535/P536 and P537/P538 to introduce a restriction site for *NdeI* directly in front of the start codon ATG and a restriction site for *PacI* directly after the stop codon TAA. The mutated sequence was cloned (*SacII* and *EcoRV*) in the Pb238 intermediate vector that contained already the *trap* 3'UTR downstream of the selection marker and the resulting plasmid was named Pb238-TRAP-NdeI/PacI. The designed DNA sequence lacking the coding region of the I domain was codon modified for *E. coli* K12 and synthesized at GeneArt (Invitrogen). Subsequently the designed sequence was cloned (*NdeI* and *PacI*) in the Pb238-TRAP-NdeI/PacI by replacing the endogenous *trap* gene. Final DNA sequences were digested for linearization (*NotI*, PbGEM-107890; *SacII* and *KpnI*, Pb238-TRAPΔI), purified and transfected into wild type parasites (*wt*) using standard protocols (*Janse et al., 2006*; *Figure 2—figure supplement 1*).

## Generation of *TRAP-I, MIC2-I, αL-I, αX-I* and *RevCharge* parasites

To generate parasite lines expressing TRAP with different I domains, bp 115 to bp 696 (581 bp; I42 to V228; 194 aa in total) of the wild type *trap* gene (*Plasmodium berghei* ANKA strain) were exchanged with sequences from the micronemal protein 2 (MIC2) of *Toxoplasma gondii* (567 bp, L75 to V263, 189 aa), the integrin CD11c/αX (552 bp, Q150 to I333, 184 aa) and the integrin CD11a/αL (531 bp, V155 to I331, 177 aa) of *Homo sapiens*. Chimeric sequences as well as the coding sequence of the wild type *trap* gene that served as a control, were codon modified for *E. coli* K12 to prevent incorrect homologous recombination with the *trap* coding sequence downstream of the I domain coding region and to avoid changes of the codon usage within the open reading frame caused by the exchanged I domain coding sequence. This enabled also simple differentiation between wild type and transgenic parasites by PCR. For the generation of *RevCharge* parasites seven mutations of non-conserved amino acids (H56E, H62E, H123E, K164Q, K165D, R195E and K202E; *P. berghei* ANKA strain) were introduced into the codon modified wild type *trap* gene. These mutations were expected to shift the surface charge at the apical side of the I domain from a pI of 9.7 to 6.8 while leaving the MIDAS intact and the structural integrity of the domain unaffected. All designed sequences were synthesized at GeneArt (Invitrogen) and cloned in the Pb238-TRAP-NdeI/PacI vector (*NdeI/PacI*) that was already used to generate the *trapΔI* line. Constructs were digested for linearization (*ScaI-HF*) and transfected using standard protocols (*Janse et al., 2006*). Transfections were performed in the negatively selected TRAP knockout line *trap(-)* as well as in the fluorescent background line *fluo* to independently generate fluorescent (*fluo*) and non-fluorescent (*non-fluo*) sets of mutants (*Figure 3—figure supplement 1*).

## Generation of isogenic parasite populations

Isogenic parasite lines were generated by serial dilution of parental populations obtained from transfections. Per transfection one mouse was infected by intraperitoneal injection of ~200 µL frozen parasites of the parental population. To increase the number of transfected parasites within infected mice pyrimethamine (0.07 mg/mL) was given within the drinking water 24 hr post injection (hpi). Donor mice were bled two to three days post injection once parasitemia reached 0.5–1%. Parasites were diluted in phosphate buffered saline (PBS) to 0.7–0.8 parasites per 100 µL and the same volume was subsequently injected into 6–10 naive mice. Parasites were allowed to grow for 8–10 days until parasitemia reached 1.5–2%. Blood of infected mice was taken by cardiac puncture (usually 600–800 µL) and used to make parasite stocks (~200 µL infected blood) and to isolate genomic DNA with the Blood and Tissue Kit (Qiagen).

## Mosquito infection

Naive mice were infected with 100–200 µL frozen parasite stocks and parasites allowed to grow for four to five days. Infected mice were either directly fed to mosquitoes or used for a fresh blood transfer of $2 \times 10^7$ parasites by intraperitoneal injection into two to three naive mice. Parasites within recipient mice were allowed to grow for further three to four days, depending on the number of exflagellation events observed. To determine the number of exflagellation events, and subsequently the number of male gametocytes, a drop of tail blood was placed on a microscope slide, covered with a coverslip and incubated for 10–12 min at room temperature (20–22°C). The number of exflagellation events was counted with a light microscope (Zeiss) and a counting grid by using 40-fold magnification with phase contrast. If at least one exflagellation event per field could be observed mice were fed to mosquitoes. Mosquitoes were starved overnight prior to the feeding to increase the number of biting mosquitoes. Per mosquito cage (approximately 200–300 female mosquitoes) two to three mice were used for feeding.

## Isolation of midgut, hemolymph and salivary gland sporozoites

To estimate the number of sporozoites, infected *Anopheles stephensi* mosquitoes were dissected on day 13, 14, 18, and 22 post infection. For the collection of hemolymph sporozoites, infected *A. stephensi* mosquitoes were cut with a needle to remove the last segment of the abdomen. Subsequently the thorax was pierced with a finely drawn Pasteur pipette filled with RPMI/PS solution. While gently pressing the pipette the haemocoel cavity was flushed with solution which dripped off the abdomen. The sporozoite solution was collected on a piece of foil and transferred into a plastic reaction tube. To determine the number of midgut sporozoites, the abdomen of infected mosquitoes was dissected with two needles and the midgut was extracted. Isolated midguts were transferred into a plastic reaction tube containing 50 µL RPMI/PS solution. For the isolation of salivary gland sporozoites, the head of infected mosquitoes was gently pulled away with a needle while fixing the mosquito in place with a second needle. Ideally the salivary glands stayed attached to the head and could be easily isolated. Salivary glands were transferred into a plastic reaction tube with 50 µL RPMI/PS. To release sporozoites, pooled midguts and salivary glands were homogenized using a plastic pestle for 2 min. To count the parasites in each sample, 5–10 µL of the sporozoite solution (1:10 dilution) was applied on a hemocytometer. Sporozoites were counted using a light microscope (Zeiss) with 40-fold magnification and phase contrast. For each counting experiment at least 10 mosquitoes were dissected. However, the number was adapted depending on the infection rate of the mosquitoes and the experiment that was performed.

## Animal experiments

To determine the prepatency of parasite lines during sporozoite transmission two different routes of infection were tested. Female C57BL/6 mice were either exposed to infected mosquitoes or infected by intravenous injection of midgut, hemolymph or salivary gland sporozoites. To infect mice by mosquito bites, mosquitoes infected with fluorescent parasite lines were pre-selected for fluorescence of the abdomen by using a stereomicroscope (SMZ1000, Nikon) with an attached fluorescence unit. Subsequently parasite positive mosquitoes were separated in cups to 10 each and allowed to recover overnight. Approximately six hours prior to the experiment mosquitoes were starved by removing salt and sugar pads. Mice were anesthetized by intraperitoneal injection of a mixture of

ketamine and xylazine (87.5 mg/kg ketamine and 12.5 mg/kg xylazine) and placed with the ventral side on the mosquito cups. Mosquitoes infected 17–24 days prior to the experiment were allowed to feed for approximately 15 min before mice were removed. During this time eyes of mice were treated with Bepanthen cream (Bayer) to prevent dehydration of the cornea. After the experiment mice were allowed to recover and tested for blood stage parasites on a daily basis by evaluation of Giemsa stained blood smears. If non-fluorescent parasite lines were tested infected mosquitoes were not pre-sorted. In these experiments midguts of mosquitoes that had taken a blood meal were dissected after the experiment and the number of midgut sporozoites was counted as described previously. Mice that were bitten by mosquitoes that contained no midgut sporozoites were excluded from the analysis.

For injections hemolymph or salivary gland sporozoites were isolated either 13–16 days (hemolymph) or 17–24 days (salivary gland) post infection. Sporozoite solutions were diluted to the desired concentration with RPMI/PS (either 10,000 or 25,000 sporozoites) and injected intravenously into the tail vein of naive mice. The presence of blood stage parasites was evaluated on a daily basis.

### In vitro gliding assay

To analyze speed and movement pattern of sporozoites, in vitro gliding assays were performed in glass-bottom 96-well plates (Nunc). Hemolymph and salivary gland sporozoites were obtained by dissecting infected *A. stephensi* mosquitoes. To free the sporozoites from salivary glands, samples were grounded with a pestle. Subsequently salivary gland samples were centrifuged for 3 min at 1,000 rpm (Thermo Fisher Scientific, Biofuge primo) to separate sporozoites from tissue. Afterwards ~40 µL of the supernatant was transferred into a new 1.5 mL plastic reaction tube and diluted with a variable volume of RPMI/PS depending on the planned number of assays and the sporozoite concentration resulting in a minimum of 50,000 sporozoites per well. For each assay about 50 µL of the sporozoite suspension was mixed with 50 µL RPMI medium containing 6% bovine serum albumin (BSA) to initiate activation. Subsequently sporozoites were allowed to attach to the bottom by centrifugation at 800 rpm for 3 min (Heraeus Multifuge S1). Using fluorescence microscopy (Axiovert 200M) with a 10x objective movies were recorded with one image every three seconds for 3 to 5 min depending on the experiment. Movies were analyzed manually using the Manual Tracking Plugin from ImageJ (*Schindelin et al., 2012*) to determine speed and trajectories of moving sporozoites. Sporozoites that were able to glide at least one full circle during a 3 min movie were considered to be productively moving while all other sporozoites were classified as non-productively moving (moving less than one circle) or non-motile.

### Live imaging

For the imaging of sporozoites within salivary glands infected mosquitoes were dissected 17–24 days post infection as described previously. Isolated salivary glands were transferred with a needle to a microscope slide containing a drop of Grace's medium (Gibco) and carefully sealed with a cover slip. Samples were imaged with an Axiovert 200M (Zeiss) using 63x (N.A. 1.3) and 10x (N.A. 0.25) objectives.

### Antibodies

For immunofluorescence assays we made use of antibodies directed against the circumsporozoite protein (CSP) and the thrombospondin related anonymous protein (TRAP). In all assays the anti-CSP antibody mAb 3D11 (*Yoshida et al., 1980*) was applied as unpurified culture supernatant of the corresponding hybridoma cell line (1:5 diluted for immunofluorescence assays). TRAP antibodies were generated against the peptide AEPAEPAEPAEPAEPAEP by Eurogenetec and the purified antibody was applied as 1:100 dilution in immunofluorescence assays. Antibodies against the same peptide have been shown previously to specifically detect TRAP by immunofluorescence and western blotting (*Ejigiri et al., 2012*). Secondary antibodies coupled to AlexaFluor 488 or Cy5 (goat anti-mouse or goat anti-rabbit) directed against primary antibodies were obtained from Invitrogen and always used as 1:500 dilution.

## Immunofluorescence assays with sporozoites

To visualize the expression and localization of TRAP in sporozoites, infected salivary glands were dissected as described previously and pooled in plastic reaction tubes containing 50 µL PBS or RPMI/PS. Afterwards salivary glands were mechanically grounded with a plastic pestle to release sporozoites from tissue. Immunofluorescence assays were performed by two different methods either fixing sporozoites in solution or on glass cover slips. To fix the parasites on glass, salivary glands were dissected in RPMI/PS and treated as described. Sporozoite solutions were transferred into 24-well plates containing round cover slips, activated with an equal volume RPMI/PS containing 6% BSA and allowed to glide for approximately 30 min at RT. Subsequently the supernatant was discarded and sporozoites were fixed with 4% PFA (in PBS) for 1 hr at RT. Fixed samples were washed three times with PBS for 5 min each. If immunofluorescence was performed on sporozoites in solution, salivary glands were dissected in PBS and treated as described previously. Sporozoite solutions were directly fixed by adding 1 mL of 4% PFA (in PBS) for 1 hr at RT. After fixation samples were washed as described for samples fixed on glass while samples in solution had to be additionally pelleted after each step by centrifugation for 3 min at 10,000 rpm (Thermo Fisher Scientific, Biofuge primo). Subsequently sporozoites were blocked (PBS containing 2% BSA) or blocked and permeabilized (PBS containing 2% BSA and 0.5% Triton X-100) over night at 4°C or for 1 hr at RT, respectively. Samples were incubated with primary antibody solutions for 1 hr at RT in the dark and subsequently washed three times with PBS. After the last washing step, samples were treated with secondary antibody solutions and incubated for 1 hr at RT in the dark. Stained samples were washed three times in PBS and the supernatant was discarded. If the immunofluorescence assay was performed in solution, sporozoite pellets were resuspended in 50 µL of remaining PBS, carefully pipetted on microscopy slides and allowed to settle for 10–15 min at RT. Remaining liquid was removed with a soft tissue and samples were covered with cover slips which had been prepared with 7 µL of mounting medium (ThermoFisher Scientific, ProLong Gold Antifade Reagent). If the immunofluorescence assay was performed on sporozoites that were fixed on glass, cover slips were removed with forceps, carefully dabbed on a soft tissue and placed on microscopy slides that had been prepared with 7 µL of mounting medium. Samples were allowed to set overnight at RT and kept at 4°C or directly examined. Images were acquired with a spinning disc confocal microscope (Nikon Ti series) with 60-fold magnification (CFI Apo TIRF 60x H; NA 1.49).

## Western blot analysis of TRAP expression

Salivary glands of infected mosquitoes were dissected in 100 µL PBS on ice and subsequently smashed with a pestle to release sporozoites. For the isolation of hemolymph sporozoites infected mosquitoes were flushed with PBS as previously described. Sporozoite solutions were kept on ice, counted using a haemocytometer and distributed to 30,000 sporozoites per reaction tube. Samples were centrifuged for 1 min with 13,000 rpm (Thermo Fisher Scientific, Biofuge primo) at 4°C to pellet sporozoites. Subsequently the supernatant was discarded, and pellets were lysed in 20 µL RIPA buffer (50 mM Tris pH 8, 1% NP40, 0.5% sodium dexoycholate, 0.1% SDS, 150 mM NaCl, 2 mM EDTA) supplemented with protease inhibitors (Sigma-Aldrich, P8340) for ≥1 hr on ice. Lysates were mixed with Laemmli buffer, heated for 10 min at 95°C and centrifuged for 1 min at 13,000 rpm (Thermo Fisher Scientific, Biofuge primo). Samples were separated on precast 4–15% SDS-PAGE gels (Mini Protein TGX Gels, Bio-Rad) and blotted on nitrocellulose membranes with the Trans-Blot Turbo Transfer System (Bio-Rad). Blocking was performed by incubation in PBS containing 0.05% Tween20% and 5% milk powder for 1 hr at RT. Afterwards, the solution was refreshed and antibodies directed against TRAP (rabbit polyclonal antibody, 1:1000 diluted) or the loading control CSP (mAb 3D11, cell culture supernatant 1:1000 diluted) were added. Membranes were washed three times (PBS with 0.05% Tween20) and secondary anti-rabbit antibodies (Immun-Star (GAR)-HRP, Bio-Rad) or anti-mouse antibodies (NXA931, GE Healthcare) conjugated to horse radish peroxidase were applied for 1 hr (1:10,000 dilution) at room temperature. Signals were detected using Super-Signal West Pico Chemiluminescent Substrate or SuperSignal West Femto Maximum Sensitivity Substrate (Thermo Fisher Scientific). After the detection of TRAP, blots were treated with stripping buffer (Glycine 15 g/L, SDS 1 g/L, Tween20 10 ml/L, pH 2,2) for 15 min prior to incubation with anti-CSP antibodies used as loading control.

## Cell lines

HepG2 cells were a gift from our virology department who had obtained the cells from ATCC. Cell line identity was regularly confirmed by SNP sequencing and visual observations of cell morphology. Cells were routinely tested for mycoplasma contamination using the MycoAlert mycoplasma detection kit (Lonza, Basel, Switzerland). We culture cells for a maximum of 10 passages.

## Liver stage development assay

Two days prior to the experiment 50,000 HepG2 cells/well were seeded in an 8-well Permanox Lab-Tek chamber slide (Nunc). On day zero salivary glands were isolated from infected female mosquitoes and collected in RPMI medium within a 1.5 mL reaction tube. Sporozoites were released by mechanically disrupting the salivary glands with a polypropylene pestle. The solution was centrifuged in a tabletop centrifuge at 1,000 rpm for 3 min at RT and the supernatant was transferred to a new 1.5 mL reaction tube. The salivary gland pellet was resuspended in 100 μL RPMI medium, smashed again using a pestle, centrifuged (1,000 rpm, 3 min, RT) and the supernatant was pooled with the first one. A 1:10 dilution of the sporozoite solution was counted in a Neubauer counting chamber and 10,000 salivary gland sporozoites were used to infect HepG2 cells per well. After 1.5 hr wells were washed twice with complete DMEM medium and HepG2 cells were allowed to grow in complete DMEM medium supplemented with 1x Antibiotic-Antimycotic (Thermo Fisher Scientific). At 24 hr and 48 hr post infection cells were fixed using ice cold methanol for 10 min at RT, followed by blocking with 10% FBS/PBS overnight at 4°C. Staining with primary antibody α-*Pb*HSP70 1:300 in 10% FBS/PBS for 2 hr at 37°C was succeeded by two washing steps with 1% FBS/PBS. Incubation with secondary antibody α-mouse Alexa Fluor 488 1:300 in 10% FBS/PBS was performed for 1 hr at 37°C. Hoechst 33342 was added and incubated for 5 min at RT followed by two washing steps with 1% FBS/PBS. The assay was mounted in 50% glycerol and sealed using a glass cover slip. Samples were imaged with an Axiovert 200M (Zeiss) microscope and subsequently analyzed using ImageJ (*Schindelin et al., 2012*). In brief, the perimeter of single liver stages was encircled, and the area measured using the internal measurement tool.

## Sporozoite invasion assay

Two days prior to the experiment 180,000 HepG2 cells/well were seeded in an 8-well Permanox Lab-Tek chamber slide (Nunc). Sporozoites were isolated as described above and 10,000 sporozoites/well were used to infect HepG2. At 1.5 hr post infection cells were washed twice with complete DMEM medium and fixed using 4% PFA/PBS 20 min at RT. Blocking was performed with 10% FBS/PBS o/n at 4°C followed by incubation with primary antibodies α-*Pb*CSP 1:100 in 10% FBS/PBS (2 hr 37°C), two washing steps with 1% FBS/PBS and incubation with secondary antibodies α-mouse Alexa Fluor 488 1:300 in 10% FBS/PBS (1 hr 37°C). After two washing steps with 1% FBS/PBS, cells were permeabilized by addition of ice cold methanol and incubation at RT for 10 min. Blocking with 10% FBS/PBS (4°C and overnight) was followed by an incubation with primary antibodies α-*Pb*CSP 1:100 in 10% FBS/PBS (2 hr at 37°C), two washing steps with 1% FBS/PBS and incubation with primary antibodies α-mouse Alexa Fluor 546 1:300 in 10% FBS/PBS (1 hr at 37°C). The assay was mounted in 50% glycerol after two washing steps with 1% FBS/PBS.

## Measuring organ-specific parasite load

To measure the parasite load of different organs C57BL/6 mice were infected by intravenous injection of 20,000 salivary gland sporozoites. At 42 hr after infection organs were harvested, homogenized and the RNA was isolated using Trizol according to the manufacturer's protocol. Isolated RNA was treated with DNase using the Turbo DNA-free Kit (Invitrogen). Subsequently RNA content of generated samples was measured using a NanoDrop and liver, intestine, spleen and lung samples were pooled in equal amounts of RNA to generate single samples for each parasite line and harvested organ. RNA pools were used to synthetize cDNA using the First Strand cDNA Synthesis Kit (Thermo Fisher Scientific). Quantitative RT-PCR was performed in triplicates on an ABI7500 (Applied Biosystems) using a 2x SYBR green Mastermix (Applied Biosystems). *Plasmodium berghei* 18S rRNA was used to quantify parasites and mouse specific GAPDH was utilized as housekeeping gene for normalization. Subsequently the ΔcT was plotted as mean of all replicates per parasite strain and harvested organ (*Schmittgen and Livak, 2008*).

## Animal work

For all experiments female 4–6 week-old Naval Medical Research Institute (NMRI) mice or C57BL/6 mice obtained from Janvier laboratories were used. Transgenic parasites were generated in the *Plasmodium berghei* ANKA background (*Vincke and Bafort, 1968*) either directly in wild type or from wild type derived strains (e.g. *trap(-)* and *fluo*). Parasites were cultivated in NMRI mice while transmission experiments with sporozoites were performed in C57Bl/6 mice only.

## Statistical analysis

Statistical analysis was performed using GraphPad Prism 5.0 (GraphPad, San Diego, CA, USA). Data sets were either tested with a one-way ANOVA or a Student's t test. A value of $p < 0.05$ was considered significant.

## Acknowledgements

We thank Christian Sommerauer and Miriam Reinig for rearing *Anopheles stephensi* mosquitoes as well as Christine Hopp for helpful discussions at the BioMalPar conference. This work was funded by the Human Frontier Science Program (RGY0071/2011), the Deutsche Forschungsgemeinschaft (DFG, German Research Foundation) – project number 240245660 – SFB 1129, NIH Grant CA 31798, and the European Research Council (ERC StG 281719). DK is currently funded by a DFG postdoctoral fellow ship (KL 3251/1–1) and is an alumnus of the Heidelberg Biosciences International Graduate School (HBIGS). FF is member of CellNetworks Cluster of excellence at Heidelberg University. We acknowledge the microscopy support from the Infectious Diseases Imaging Platform (IDIP) at the Center for Integrative Infectious Disease Research.

## Additional information

### Funding

| Funder | Grant reference number | Author |
| --- | --- | --- |
| Human Frontier Science Program | RGY0071/2011 | Friedrich Frischknecht |
| European Research Council | ERC StG 281719 | Friedrich Frischknecht |
| German Research Foundation | 240245660 - SFB 1129 | Friedrich Frischknecht |
| National Institutes of Health | CA 31798 | Timothy A Springer |
| German Research Foundation | KL 3251/1-1 | Dennis Klug |

The funders had no role in study design, data collection and interpretation, or the decision to submit the work for publication.

### Author contributions

Dennis Klug, Conceptualization, Data curation, Formal analysis, Funding acquisition, Investigation, Visualization, Methodology, Writing - original draft, Writing - review and editing; Sarah Goellner, Jessica Kehrer, Julia Sattler, Léanne Strauss, Data curation, Formal analysis, Investigation, Writing - review and editing; Mirko Singer, Resources, Investigation, Methodology, Writing - review and editing; Chafen Lu, Conceptualization, Resources, Writing - review and editing; Timothy A Springer, Conceptualization, Data curation, Formal analysis, Supervision, Funding acquisition, Writing - original draft, Project administration, Writing - review and editing; Friedrich Frischknecht, Conceptualization, Supervision, Funding acquisition, Writing - original draft, Project administration, Writing - review and editing

### Author ORCIDs

Dennis Klug [ID] https://orcid.org/0000-0002-9108-454X
Sarah Goellner [ID] https://orcid.org/0000-0002-3300-4273
Jessica Kehrer [ID] https://orcid.org/0000-0001-5084-3485

Léanne Strauss https://orcid.org/0000-0002-0680-593X
Mirko Singer https://orcid.org/0000-0002-5757-2750
Timothy A Springer https://orcid.org/0000-0001-6627-2904
Friedrich Frischknecht https://orcid.org/0000-0002-8332-6668

## Ethics

Animal experimentation: All animal experiments were performed according to the German Animal Welfare Act (Tierschutzgesetz) and executed following the guidelines of the Society of Laboratory Animal Science (GV-SOLAS) and of the Federation of European Laboratory Animal Science Associations (FELASA). All experiments were approved by the responsible German authorities (Regierungspräsidium Karlsruhe, G-134/14 and G266/16). All interventions were performed under ketamine/xylazine anesthesia, and every effort was made to minimize suffering.

## Decision letter and Author response

Decision letter https://doi.org/10.7554/eLife.57572.sa1
Author response https://doi.org/10.7554/eLife.57572.sa2

## Additional files

### Supplementary files

• Supplementary file 1. Primer sequences. Primers used for the generation and genotyping of the parasite lines presented in this study.

• Supplementary file 2. Amino acid sequences of the TRAP variants expressed by the parasite lines *TRAP-I, MIC2-I, αX-I, αL-I, and RevCharge*. Shown are the sequences of each TRAP replacement. Residues that are part of the extendable ß-ribbon are written in green, residues that form the remainder of the I domain are written in red, residues of the thrombospondin domain are written in orange, and the remaining native residues of *Pb*TRAP are written in black. Residues written in blue were introduced into wild type *Pb*TRAP to generate a more negative charge on the portion of the I domain surface surrounding the MIDAS in the *RevCharge* mutant. Residues written in white on a black background were mutated to create a better fitting of the exchanged portion of the I domain with the N- and C-terminal segments of the *Pb*TRAP I domain/extendable ß-ribbon. The calculated pI of the I domain region is shown in parentheses.

• Transparent reporting form

### Data availability

All data generated or analysed during this study are included in the manuscript and supporting files.

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
