## [Decision Letter]

**Acceptance summary:**

The thrombospondin related anonymous protein (TRAP) is an important adhesin expressed by the sporozoite stage of the malaria parasite, required for sporozoite motility and invasion of the salivary gland in the mosquito and of the liver in the mammalian host. To gain insight into the contribution of integrin I-domain of TRAP, the authors performed deletion as well as creative swap between Plasmodium, Toxoplasma and mammalian integrin I-domains. These technically challenging experiments established that the I domains have the versatile capacity to confer adhesive properties to TRAP both in vertebrate and arthropod hosts.

**Decision letter after peer review:**

[Editors’ note: the authors submitted for reconsideration following the decision after peer review. What follows is the decision letter after the first round of review.]

Thank you for submitting your work entitled "Cationic charge and polyspecificity of an integrin domain regulates infectivity of malaria parasites" for consideration by *eLife*. Your article has been reviewed by three peer reviewers, and the evaluation has been overseen by a Reviewing Editor and a Senior Editor. The reviewers have opted to remain anonymous.

Our decision has been reached after consultation between the reviewers. Based on these discussions and the individual reviews below, we regret to inform you that your work will not be considered for publication in *eLife* at this stage.

As you will see, the three reviewers are broadly supportive of this work, which addresses an important topic. However, they were especially concerned that the conclusions drawn from the a-L and a-X swapped mutants are not sufficiently supported by the data. Importantly they raised three main issues that will take some time to address (more than two months) and in consequence, we have together opted for a rejection with a strong encouragement for re-submission. If you are able to address these issues we will consider a newly submitted form of this paper that we will treat as a revised manuscript. If you choose this course of action please submit a separate cover letter detailing the changes you have made.

Main points to be addressed:

1) Establish if the two human integrin I-domain swaps have the same phenotype as the trap null allele. At the moment the authors imply that they are strong hypomorphs but two reviewers point out there is no side-by-side quantitative comparison with the null. If the human integrin I-domain swaps do phenocopy the null then the wording in the manuscript must be changed significantly.

2) Make a direct link between I-domain ligand binding and phenotype rather than implying this is the reason for the observed effects. The suggestion of performing the gliding assays on different substrates, and especially on known ligands of the swapped I-domains, is a good one and would be informative.

3) Establish that the I-domains retain their folding and binding functions in the context of the chimeric trap.

Reviewer #1:

In this study, Klug and colleagues report on the role of the integrin I-domain of TRAP, an important adhesin expressed by the sporozoite stage of the malaria parasite. Previous studies have shown that TRAP is required for sporozoite motility and invasion of the salivary gland in the mosquito and of the liver in the mammalian host. A metal ion-dependent adhesion site (MIDAS) within the I-domain of TRAP is known to play a role in sporozoite infectivity, however the functional and adhesive properties of the I-domain remain undefined. More generally, the mechanisms of sporozoite infection are poorly understood at the molecular level, so this study addresses an important topic.

To investigate the function of TRAP I-domain, the authors generated a number of mutants in *P. berghei* and performed robust phenotype analysis including gliding motility assays and analysis of sporozoite infectivity in the mosquito (invasion of salivary gland) and in the mouse (invasion of the liver). The work is very well performed and includes a large body of data, combining genetic approaches, in vitro and in vivo experiments. The paper is clearly written and the conclusions drawn from the results generally justified.

The data convincingly show that:

a) Sporozoites expressing a truncated TRAP lacking the I-domain are non-motile and do not invade the mosquito salivary glands and the mouse liver.

b) Replacement of TRAP I-domain with the I-domain from Toxoplasma MIC2 results in normal invasion of salivary glands and normal sporozoite infectivity in mice, although gliding motility is reduced.

c) Replacement of TRAP I-domain with the I-domain from human aX or aL integrin results in a severe reduction of sporozoite infectivity and gliding motility, although residual infectivity was observed in both mosquito and mouse.

d) Introduction of 7 point mutations in the I-domain of TRAP, to replace positively charged residues (H/K/R) by negatively charged residues (E/D), results in normal invasion of mosquito salivary glands yet reduced motility and slightly reduced infectivity in the mouse.

Based on the presented data, the authors conclude that the adhesive properties of the I-domain of TRAP can be partially fulfilled by the I-domain from other organisms, and that the cationic charge and the polyspecificity of the I-domain regulates sporozoite infectivity. However, the main limitation of the study is that adhesion and ligand binding were not directly assessed, so the aforementioned conclusions are based on indirect evidence or speculation. It is important to experimentally address the adhesive properties of the chimeric TRAP proteins expressed in the mutant parasites.

Major points

1) The authors conclude that the observed phenotypic defects are due to altered ligand binding properties of mutated TRAP proteins. However this was not addressed experimentally. Other modifications in the mutants, such as an alteration of the conformation of TRAP, could also contribute to the observed phenotype, unrelated with TRAP-I adhesive properties. In the absence of direct experimental evidence, it seems too speculative to conclude that the polyspecificity of the I-domain regulates sporozoite infectivity. The data clearly shows that MIC2-I almost completely complement TRAP-I functions, whilst aX-I and aL-I are poorly functional in this context. TRAP-I is basic (like aX-I), but MIC2-I is acidic (like aL-I). The data therefore show no correlation between infectivity and charges, and thus do not support an important role of the I-domain charges. Other factors may explain why MIC2-I is functionally superior to the I-domain from humain integrins, including the size of the domain size (MIC2 is closest to TRAP: 194 aa for PbTRAP-I; 189 aa for MIC2-I; 184 aa for aX-I; 177 aa for aL-I). The authors cannot exclude that a smaller I-domain may affect the conformation of TRAP protein, including other parts of the proteins such as the β ribbon. Surprisingly, the data also show no direct correlation between gliding motility and infectivity, since MIC2 and RevCharge mutants show severely reduced motility yet invade almost normally the mosquito and the mouse. This questions the relevance of the gliding assay. How do sporozoites with reduced gliding motility invade mosquito salivary glands and mouse cells?

Maybe a simple way to address these issues would be to perform gliding assays not merely on glass slides, as performed here, but on specific substrates, including known ligands of MIC2, aX or aL integrins. This would strengthen the conclusion that the effects of the I-domain modifications are really due to changes in the adhesional properties of the chimeric proteins. For example, do sporozoites expressing the aL-I domain recover gliding motility on ICAM1?

2) The phenotype of the RevCharge mutant should be more thoroughly investigated to better document a defect in sporozoite infectivity for the liver. Figure 6F shows a partial reduction of parasite fitness in the mouse, which could be quantified more precisely. In vitro assays in HepG2 cultures, as performed with the MIC2 mutant, should be included. This is important to strengthen or invalidate the hypothesis that charges in the I-domain play an important role in sporozoite infectivity.

3) The authors checked by immunofluorescence that TRAP is expressed in the mutant sporozoites. However, they should also provide evidence that the level of expression of TRAP is unaffected in the mutants, ideally using Western Blot as shown for the RevCharge mutant. This is particularly important as sequences codon-optimized for *E. coli* were used, which may differentially affect the level of expression of the proteins. Why using *E. coli* optimized sequences?

Reviewer #2:

The manuscript by Klug et al. investigates the biochemical and functional properties of I-domains by genetically replacing the I-domain within an important malaria parasite ligand called TRAP with I domains taken from another parasite (*T. gondii*) ligand, and two from human integrins. Plasmodium parasites lacking either the entire TRAP locus or just the I-domain of TRAP do not move, or invade the vector salivary glands. The authors find that the exogenous I domains from different sources can, to varying degrees, complement TRAP function resulting in the partial rescue of some motility, and ability to invade. The authors notice that the surface charges of TRAP in the rodent parasite *P. berghei* are cationic and to determine if this is important for function, mutate these residues to acidic amino acids and show again that the functions of TRAP are impaired.

The manuscript is very nicely put together with high quality data, well-prepared figures, and the writing is clear. Overall, I found this a rather strange study where the authors have used the phenotyping available in the *Plasmodium berghei* rodent infection model to investigate general properties of I domains from different proteins and organisms.

Major points

The authors take several precautions to try to ensure the I-domains retain their function when transplanted in the Plasmodium TRAP sequence by taking structural considerations into account and showing that the proteins are detectable using anti-TRAP antibodies in the transgenic parasites. A key point in my mind is whether the human integrin I-domains really do complement the function of the I-domain in TRAP. In the manuscript, the authors show that the I-domain mutant is a null (Figure 2B) and that the human integrin I -domain replacement parasites are very strong hypomorphs (Figures 3F, 4A); however, there does not appear to be a direct side-by-side comparison – e.g. injecting 25,000 HL sporozoites of the TRAP I-domain deletion alongside the integrin replacements in Figure 3—figure supplement 3B, for example. Also, are the authors able to demonstrate that the inserted I-domains retain their functions from their original proteins – is there a known ligand for the inserted integrin I-domains?

For the revCharge mutant, the authors imply that the loss of motility and ability to infect mice is due to the alteration in I-domain ligand binding, but could not an alternative explanation be that a proportion of the protein no longer folds correctly or that interactions with other parasite proteins necessary for full function are altered? The authors suggest the role of the I-domain is to interact with the host substrate, but these are not defined in the manuscript making it conceptually a rather large leap to link the mutations made in the protein to the phenotypes seen in the parasite.

Reviewer #3:

In this study the authors swap integrin "I" domains from MIC2 and mammalian integrins with the I-domain of TRAP, the key motility adhesin of malaria sporozoites. Though the domain structure of the extracellular portion of TRAP has long been known to include an integrin-like I-domain, previous mutational studies of this domain were not that informative and the idea to perform domain-swapping experiments between Plasmodium, Toxoplasma and mammalian integrin I-domains is creative and potentially informative. Overall, the MIC2 swapped TRAP and the charge-altered TRAP are the most informative data while the conclusions drawn from the a-L and a-X swapped mutants are not justified by the data shown.

1) The human integrin I-domain swapped mutants essentially behave like the TRAP KO. There is an elegant explanation in the Discussion as to why the a-L and a-X domains may not open and close in the same way as the I-domains of TRAP and MIC2. However, in the face of no data to support this or another hypothesis, it is equally plausible that there is some significant misfolding on the TRAP with a mammalian I domain. It would be important to show that this is not the case. Since there are commercially available antibodies to human integrins, can they be used, especially ones specific for conformational epitopes, to confirm the overall folding of the a-L and a-X domains.

2) The data shown for the a-L and a-X mutants do not support the conclusions. These mutants have less than 200 sporozoites in their salivary glands, the a-L mutant is not motile and not infectious while the a-X mutant is minimally motile and very poorly infectious after IV inoculation. These data look like the TRAP KO (Sultan et al., 1997), where they observed 100 to 200 sporozoites in the salivary glands and had a few infected mice when sporozoites were inoculated IV. Given this, how can they state: "Parasites expressing TRAP with I-domains from other organisms including humans can have nearly intact motility and invasion”. Or "…I-domain of TRAP is required to mediate adhesional properties which can be partially preserved when the native I-domain is replaced by I-domains from human integrins…". These statements are in the Introduction and Abstract but additionally, a central theme of the Discussion is that the a-L and a-X mutants are sufficiently intact to draw meaningful conclusions.

3) TRAP expression in mutant lines is verified by immunofluorescence that is not quantified. Since decreased levels of mutant TRAP could have phenotypes due to the mutation or to expression level, its critical to have western blot verification of mutant TRAP expression compared to WT TRAP expression, in all lines analyzed.

[Editors’ note: further revisions were suggested prior to acceptance, as described below.]

Thank you for submitting your article "Plasmodium adhesin TRAP retains function when its ligand-binding domain replaced by evolutionarily distant I domains" for consideration by *eLife*. Your article has been reviewed by three peer reviewers, and the evaluation has been overseen by a Reviewing Editor and Dominique Soldati-Favre as the Senior Editor. The reviewers have opted to remain anonymous.

The reviewers have discussed the reviews with one another and the Reviewing Editor has drafted this decision to help you prepare a revised submission.

Specifically, the interpretation of the human I domain swaps needs to be carefully worded to avoid over-interpreting the data, which seems to be the case here.

Summary:

In this study, Klug and colleagues report on the role of the integrin I-domain of TRAP, an important adhesin expressed by the sporozoite stage of the malaria parasite. Previous studies have shown that TRAP is required for sporozoite motility and invasion of the salivary gland in the mosquito and of the liver in the mammalian host. The authors swap integrin "I" domains from the *Toxoplasma gondii* ligand MIC2 and mammalian integrins with the I-domain of TRAP, the key motility adhesin of malaria sporozoites. Though the domain structure of the extracellular portion of TRAP has long been known to include an integrin-like I-domain, previous mutational studies of this domain were not that informative and the idea to perform domain-swapping experiments between Plasmodium, Toxoplasma and mammalian integrin I-domains is creative and potentially informative. Overall, the MIC2 swapped TRAP and the charge-altered TRAP are the most informative data while the conclusions drawn from the a-L and a-X swapped mutants are not justified by the data shown.

One major reservations of the previous iteration of this study was that the conclusions drawn from the parasites in which the TRAP I-domain was replaced by the human a-L and a-X integrin domains were not justified by the data shown and this has not been corrected in the revised manuscript. Indeed, salivary gland sporozoite numbers of the a-L and a-X mutants are in the same range as the TRAPΔI and TRAP KO parasites, with only the prevalence of infection somewhat increased in the a-X parasites. Infectivity studies demonstrate that the a-L parasites are similar to the TRAPΔI and TRAP KO parasites. With numerous careful experiments they that show the a-X parasites are more infectious than the a-L, TRAPΔI and TRAP KO parasites, and this interesting finding suggests that the charge on a-X may enable the mutant TRAP to have some minimal activity. However, their data show that the a-X sporozoites are about 100-fold decreased in their infectivity compared to controls. It has been previously demonstrated that a one day delay in patency is equivalent to a 10-fold decrease in infection. When 10,000 hemolymph sporozoites are inoculated IV, 100% of control mice are infected with a prepatent period of 4 days. With the a-X parasites only 50% of mice are infected and the prepatent period is 6 days. Thus, even if one ignores the fact that half the mice were not infected, the ones that did get infected had a 100-fold lower functional inoculum.

Characterization of this significant attenuation of infectivity of the a-X parasites in mosquito and mammalian hosts as "partial restoration" of TRAP function is misleading. Indeed given the phenotype they describe, it is clear that the a-X parasites cannot be naturally transmitted for a single life cycle. Thus, the paper should focus on the interesting phenotypes of the MIC2 and reverse charge sporozoites, and only compare the a-X parasites to the a-L and TRAP KO parasites where there is a small marginal increase in infectivity supporting the importance of charge in the TRAP I-domain.

The following statements in the paper should be deleted, or changed to only include the MIC-2 mutant.

Title:

The current title implies that human I domains confer some functionality to TRAP which, given that these mutants have salivary gland sporozoite counts in the range of the TRAP KO, and infectivity levels that are 100-fold decreased, is misleading.

Abstract:

"…its replacement with I domains from Toxoplasma MIC2 or the human integrin αX-subunit substantially or partially restores these functions".

"Results with TRAP, MIC2 and aX-I domains show remarkable interchangeability in their ability to engage with ligands in divergent tissues in arthropoa and chordata".

"…results suggest that apicomplexan parasites appropriated I domains in part for their ability to recognize structurally diverse ligands"

"…permit TRAP to function as an adhesin with diverse ligands in both vertebrate and arthropod hosts"

The ability to engage with ligands is not addressed here, so wording should be revised. Also, the aX-I domain is poorly functional in TRAP, so "remarkable interchangeability" is not correct in this case.

Introduction:

"Parasites expressing TRAP with I domains from other organisms, including humans show partially rescued motility and invasion."

Results:

"This could hint that the charge of αX could compensate for the long evolutionary distance between apicomplexa and mammals."

Discussion:

"It was therefore remarkable that TRAP function was largely or partially restored by replacement of its I domain by homologues from Toxoplasma gondi and *Homo sapiens* with only 28% and 18% amino acid sequence identity, respectively."

"TRAP function was largely or partially restored by replacement of its I domain". This is true for Toxoplasma MIC2 domain but not human integrin I domains.

"…immunofluorescence showed normal localization in sporozoites, suggesting that they were properly folded".

The immunofluorescence data provide no information on the folding of TRAP.

"Infectivity of hemolymph αX-I and αL-I sporozoites in mice was decreased at least 4-fold and 15-fold, respectively." Infectivity in the a-L sporozoites was similar to the TRAP KO, and in the a-X sporozoites was 100-fold decreased compared to controls.

"The findings that I domains from organisms that encounter none of the same ligands as TRAP can replace its function in mediating salivary gland invasion and infection of mice strongly suggest that the I domain functions to provide traction for motility and cell invasion rather than tissue tropism." This should specify that they are referring to replacement with MIC2.

"…the I domain of TRAP is required for gliding motility and invasion by sporozoites of mosquito salivary glands and mammalian liver, without a requirement for evolved specificity".

The data with the human I domains show that any I domain cannot confer function in TRAP, so this statement is not correct.

The authors attempted to analyze the motility of mutant sporozoites on various substrates. They indicate that they "were not able to measure a significant effect of a specific ligand".

However, Figure 1 shows that aX-I and aL-I activated on BSA are mostly floating, while on the other substrates sporozoites are mostly attached or display unproductive motility. This could be an indication that the human I-domains retained some adhesive properties. However, it would be useful to have the same type of data with the I domain-deleted mutant, and the threshold, as indicated in A, should also be shown on the other panels.

---

## [Author Response]

[Editors’ note: the authors resubmitted a revised version of the paper for consideration. What follows is the authors’ response to the first round of review.]

As you will see, the three reviewers are broadly supportive of this work, which addresses an important topic. However, they were especially concerned that the conclusions drawn from the a-L and a-X swapped mutants are not sufficiently supported by the data. Importantly they raised three main issues that will take some time to address (more than two months) and in consequence, we have together opted for a rejection with a strong encouragement for re-submission. If you are able to address these issues we will consider a newly submitted form of this paper that we will treat as a revised manuscript. If you choose this course of action please submit a separate cover letter detailing the changes you have made.

Thank you for the constructive and encouraging evaluation of our manuscript. Please find in the following our comments describing the changes in the manuscript we introduced and the additional experiments we performed to address the comments from the reviewers. Please note that due to the extensive changes and the suggestion by reviewer 3 to remove part of the supplement two authors who participated in the deleted work were removed from the author list and two authors changed position reflecting their respective contributions since the last submission. In total we added data from over 30 mosquito cage infections to improve the infection data of some of the mutants and to provide adequate Western blots for all mutants, who did not enter the salivary glands from hemolymph sporozoites.

Main points to be addressed:1) Establish if the two human integrin I-domain swaps have the same phenotype as the trap null allele. At the moment the authors imply that they are strong hypomorphs but two reviewers point out there is no side-by-side quantitative comparison with the null. If the human integrin I-domain swaps do phenocopy the null then the wording in the manuscript must be changed significantly.

Thank you for this suggestion, which helped us clarifying a discussion amongst the authors when writing the manuscript. We now performed a side by side comparison with the null mutant and the I-domain deletion as recommended (Table 2 of the manuscript). The new results show that the aX line partially rescues TRAP function, as sporozoite injection led to the infection of around 50% of all mice (11/20) that received a sporozoite injection while no mice became blood stage patent if infected with the I-domain deletion (0/14) or the TRAP knockout (0/10); note that Table 3 summarizes these data for injections with hemolymph sporozoites. Only one mouse (from 20) could be infected with the aL line. Therefore we agree with the reviewers that the aL mutant is essentially not rescuing TRAP functions and changed the wording across the manuscript accordingly. The new data were incorporated into Table 2 and a Table 3 was added that sums up the key data from Table 2 for faster direct comparison between the mutant lines.

2) Make a direct link between I-domain ligand binding and phenotype rather than implying this is the reason for the observed effects. The suggestion of performing the gliding assays on different substrates, and especially on known ligands of the swapped I-domains, is a good one and would be informative.

Ligand binding by TRAP is indeed a key question for sporozoite biology. In the literature several potential ligands of TRAP are described such as proteoglycans and Fetuin-A on hepatocytes, saglin on salivary glands and an integrin in the skin. This broad range of potential ligands indicates already that TRAP interacts promiscuously with different proteins on different tissues even if the interaction might be specific in each case. It is also important to point out that these putative ligands are only important for TRAP mediated invasion but not needed for TRAP mediated motility although the data for the integrin in the skin suggest a minor effect on in vivo motility (Dundas et al., 2018).

While productive movement is dependent on TRAP and, as we show in this manuscript, also requires the presence of the I-domain within TRAP no specific TRAP-ligand is needed to activate sporozoites. In fact sporozoites do glide even in RPMI medium that contains only amino acids without protein additives or even in PBS with just added glucose. Hence, addition of a protein like bovine serum albumin (BSA) boosts activation and/or adhesion of sporozoites and increases the percentage of gliding parasites. However, other added proteins show similar effects. Therefore testing ligand specificity of TRAP in a motility assay is unfortunately not trivial; a rather disappointing observation that cost us many hairs over the years.

To illustrate this, we include Author response image 1. This shows results from gliding assays within 96-well plates coated with different proteins of the extracellular matrix (laminin, fibronectin and collagen), ICAM-I, a known ligand of the aL-I domain and heparin, a ligand of the aX-I domain. We tested also the gliding behaviour of sporozoites of all four lines in the presence of seven differently charged heptapeptides as well as in the presence of BSA (positive control) and pure RPMI medium (negative control). The results of these experiments are compiled below.

**Author response image 1. sa2fig1:** In vitro gliding motility of TRAP-I, MIC2-I, αX-I and αL-I sporozoites on different substrates. Sporozoites were allowed to glide in wells coated with heparin, laminin, fibronection, collagen and two different concentrations of ICAM-I (10 µg/mL and 20 µg/mL); see Author response table 1. In addition gliding motility was tested in the presence of seven differently charged heptapeptides as well as in 3% BSA and in RPMI(P/S) without additives. (**A**) Gliding motility of TRAP-I salivary gland sporozoites. The red line indicates the baseline of productive motility observed in the negative control RPMI(P/S). Per condition 184 to 415 sporozoites were analyzed. (**B**) Gliding motility of MIC2-I salivary gland sporozoites. Per condition 111 to 174 sporozoites were analyzed. (**C**) Gliding motility of MIC2-I hemolymph sporozoites. Per condition 59 to 200 sporozoites were analyzed. (**D**) Gliding motility of αX-I hemolymph sporozoites. Per condition 97 to 366 sporozoites were analyzed. (**E**) Gliding motility of αL-I hemolymph sporozoites. Per condition 167 to 240 sporozoites were analyzed. (**F**) Classification of movement pattern according to Figure 2—figure supplement 2 of the manuscript. (**G**) Sequence and charge of tested heptapeptides. The CPPpred Score (https://omictools.com/cpppred-tool) gives the probability of the peptide to be membrane permeable (<0.5; unlikely to cross membranes, >0.5; likely to cross membranes). Note that the assay was not performed with TRAP-I hemolymph sporozoites because we did not observe any specific activation of gliding motility for αX-I and αL-I hemolymph sporozoites in any condition. All assays were performed in duplicates with sporozoites obtained from two mosquito feeding experiments. nd: not determined.

Evaluating substrate specificity of *TRAP-I*, *MIC2-I*, *αX-I* and *αL-I* sporozoites:

We tested gliding motility in vitro for sporozoites of the four parasite lines TRAP-I, TRAP-MIC2, TRAP-αX and TRAP-αL in 96-well plates coated with heparin, ICAM-I, laminin, fibronectin and collagen. We included also a library of seven differently charged heptapeptides that we had available in our laboratory. As positive control we made use of 3% bovine serum albumin (BSA) and as negative control we tested RPMI medium mixed with penicillin and streptomycin that was used as base in all conditions. Hemolymph and salivary gland sporozoites were obtained by dissecting infected *A.stephensi* mosquitoes as described in Materials and methods. In brief, salivary glands were dissected and transferred into a plastic reaction tube containing 50 µL RPMI/PS. Salivary glands were ground with a pestle and centrifuged for 3min at 1000rpm at room temperature to separate freed sporozoites from mosquito tissue. Subsequently approximately 40µL of the supernatant was transferred into a new reaction tube. For the collection of hemolymph sporozoites infected *A. stephensi* mosquitoes were cut with a needle to remove the last segment of the abdomen. Subsequently the thorax was pierced with a finely drawn Pasteur pipette filled with RPMI/PS solution. While gently pressing the pipette the haemocoel cavity was flushed with solution which dripped off the abdomen. The sporozoite solution was collected on a piece of foil and transferred into a plastic reaction tube. Sporozoite concentration of salivary gland and hemolymph sporozoites was subsequently calculated using a hemocytometer. Afterwards sporozoites were diluted with RPMI/PS to achieve a volume of 50 µL per tested condition containing not less than 5000 sporozoites. Depending on the condition 50µL of sporozoite suspension was mixed with either 50µL RPMI containing 6% bovine serum albumin (BSA) or with 50µL peptide solution for activation. Peptides were ordered as solid substrates and solubilized in water to generate stocks with a final concentration of 10mg/mL. Using RPMI/PS the stock solution was diluted to an initial concentration of 2mg/mL to achieve a final dilution of 1mg/mL after addition of sporozoites. Alongside to BSA and peptides, six different coatings were tested for sporozoite motility. Coated wells were loaded with 50 µL sporozoite solution that was mixed with an equal volume of RPMI/PS. As a negative control sporozoites were allowed to glide in an uncoated well in RPMI/PS without any addition of peptides or BSA. To ensure the same conditions for each gliding assay, samples were prepared in 3min intervals. To allow sporozoites to attach to the optical bottom, samples were centrifuged at 800rpm for 3min before imaging. Using fluorescent microscopy (Axiovert 200M; 10x objective with NA 0.25) videos were recorded with three seconds per frame for 3min. Subsequently data were analyzed using ImageJ.

**Author response table 1. resptable1:** Coating protocols used to test gliding motility of sporozoites on different substrates. Gliding assays were performed in 96-well plates and wells were coated with heparin, ICAM-I, laminin, fibronectin and collagen according to the following protocols (Bilsland, Diamond and Springer, 1994; Gao et al., 2011).

Coating agent	Concentration	Protocol
Heparin	100 U/µL	Heparin (stock: 25000 U/µL) was diluted in Laminin buffer (150 mM NaCl, 50 mM TRIS, pH 7.4) to 100 U/µL. Coating procedure: Per well 150 µL heparin solution was added and incubated overnight at 4°C. Before the gliding assay was started wells were washed once with PBS.
ICAM-I	10 µg/mL	ICAM-I (stock: 2 mg/mL) was diluted in PBS to 10 or 20 µg/mL. Coating procedure: Per well 150 µL of the final solution was added and incubated at 4°C overnight. Before the gliding assays were started wells were washed once with PBS.
ICAM-I	20 µg/mL
Laminin	25 µg/mL	Laminin (stock: 1 mg/mL) was diluted in Laminin buffer to 25 µg/mL. Coating procedure: Wells were washed with 70 % EtOH to increase hydrophilicity. Subsequently wells were washed three times with H_2_O to remove remnants of EtOH. Per well 200 µL laminin solution was added and incubated at room temperature for one hour. Before the gliding assay was started wells were washed once with PBS.
Fibronectin	50 µg/mL	Fibronectin (stock: 1 mg/mL) was diluted in Laminin buffer to 50 µg/mL. Coating procedure: As above.
Collagen	2.5 µg/mL	Collagen (stock: 1 mg/mL) was diluted in 0.2 M acetic acid to 2.5 µg/mL. Coating procedure: As above.

In contrast to the intuitive hypothesis that known ligands of the human integrin I-domains should activate sporozoite motility we were not able to measure a significant effect of a specific ligand. aL-I sporozoites and aX-I sporozoites isolated from the hemolymph were either not able to glide productively or showed gliding at very low percentages (1/100-200 sporozoites). Control sporozoites from salivary glands showed about 50% of actively gliding sporozoites in BSA while already about 10% of sporozoites were gliding in the negative control (RPMI medium only). In other conditions sporozoite activation was lower than with BSA but mostly above 10%. We couldn’t observe any significant differences in activation between different compounds other than BSA.

While these results were disappointing at first they agree with published observation of sporozoite activation (e.g. see Perschmann et al., Nano Letters 2011).

3) Establish that the I-domains retain their folding and binding functions in the context of the chimeric trap.

We agree with the reviewers that this is an important question. Naturally, verification of protein folding in vivo is a difficult task. We already considered the suggestion of reviewer 3 to do immunofluorescence with I-domain specific antibodies during our studies. Indeed there are antibodies targeting integrins available that target the I-domain of human integrin aX and aL available. Yet we feel that a proper positive and negative control would necessitate further parasite lines such as a TRAP-aX line with a targeted mutation that disrupts the conformation or epitope.

As an alternative to generating more lines that could be less infective as the ones at hand, we performed more infection experiments. These showed that aX can rescue sporozoite infectivity to mice at a rather high level, while aL could not (only one infected mouse). In addition, we performed additional experiments with the RevCharge mutant, which revealed an interesting functionality: the RevCharge mutant infects salivary glands normally, yet it shows a severe impairment in gliding motility and infectivity of mice. This suggests that the folding of the protein must be correct yet, it cannot fully function in the mammalian host. In the aX line we observed invasion of salivary glands at very low rates as well as little (but clearly detectable) ability to actively glide in circles (Video 2). However, injected hemolymph sporozoites caused infections in more than 50% of all treated mice (9/16; Tables 2 and 3). This suggests that the I-domain is partly functional in this mutant. Hence, one domain (RevCharge) allows salivary gland invasion, while both allow some degree of mice infection. All these observations document a restored function of TRAP that could not be observed if the I-domain is not at least partially functional. We performed also western blot experiments with hemolymph sporozoites of all lines that could not enter the salivary gland showing that TRAP is expressed at comparable levels – we found similar results in midgut sporozoites but these experiments suffered from degradations and are thus not shown. These data are now added to Figure 3.

Taken together, the data now suggest that TRAP is functionally folded in the TRAP-I, MIC-I, TRAP-RevCharge and the aX-I line while we cannot be sure if this is true for the aL line. We therefore changed the wording across the manuscript to point out that folding of the I-domain could have affected the phenotype observed in aL parasites.

Reviewer #1:In this study, Klug and colleagues report on the role of the integrin I-domain of TRAP, an important adhesin expressed by the sporozoite stage of the malaria parasite. Previous studies have shown that TRAP is required for sporozoite motility and invasion of the salivary gland in the mosquito and of the liver in the mammalian host. A metal ion-dependent adhesion site (MIDAS) within the I-domain of TRAP is known to play a role in sporozoite infectivity, however the functional and adhesive properties of the I-domain remain undefined. More generally, the mechanisms of sporozoite infection are poorly understood at the molecular level, so this study addresses an important topic.To investigate the function of TRAP I-domain, the authors generated a number of mutants in P. berghei and performed robust phenotype analysis including gliding motility assays and analysis of sporozoite infectivity in the mosquito (invasion of salivary gland) and in the mouse (invasion of the liver). The work is very well performed and includes a large body of data, combining genetic approaches, in vitro and in vivo experiments. The paper is clearly written and the conclusions drawn from the results generally justified.The data convincingly show that:a) Sporozoites expressing a truncated TRAP lacking the I-domain are non-motile and do not invade the mosquito salivary glands and the mouse liver.b) Replacement of TRAP I-domain with the I-domain from Toxoplasma MIC2 results in normal invasion of salivary glands and normal sporozoite infectivity in mice, although gliding motility is reduced.c) Replacement of TRAP I-domain with the I-domain from human aX or aL integrin results in a severe reduction of sporozoite infectivity and gliding motility, although residual infectivity was observed in both mosquito and mouse.d) Introduction of 7 point mutations in the I-domain of TRAP, to replace positively charged residues (H/K/R) by negatively charged residues (E/D), results in normal invasion of mosquito salivary glands yet reduced motility and slightly reduced infectivity in the mouse.Based on the presented data, the authors conclude that the adhesive properties of the I-domain of TRAP can be partially fulfilled by the I-domain from other organisms, and that the cationic charge and the polyspecificity of the I-domain regulates sporozoite infectivity. However, the main limitation of the study is that adhesion and ligand binding were not directly assessed, so the aforementioned conclusions are based on indirect evidence or speculation. It is important to experimentally address the adhesive properties of the chimeric TRAP proteins expressed in the mutant parasites.Major points1) The authors conclude that the observed phenotypic defects are due to altered ligand binding properties of mutated TRAP proteins. However this was not addressed experimentally. Other modifications in the mutants, such as an alteration of the conformation of TRAP, could also contribute to the observed phenotype, unrelated with TRAP-I adhesive properties. In the absence of direct experimental evidence, it seems too speculative to conclude that the polyspecificity of the I-domain regulates sporozoite infectivity. The data clearly shows that MIC2-I almost completely complement TRAP-I functions, whilst aX-I and aL-I are poorly functional in this context. TRAP-I is basic (like aX-I), but MIC2-I is acidic (like aL-I). The data therefore show no correlation between infectivity and charges, and thus do not support an important role of the I-domain charges. Other factors may explain why MIC2-I is functionally superior to the I-domain from humain integrins, including the size of the domain size (MIC2 is closest to TRAP: 194 aa for PbTRAP-I; 189 aa for MIC2-I; 184 aa for aX-I; 177 aa for aL-I). The authors cannot exclude that a smaller I-domain may affect the conformation of TRAP protein, including other parts of the proteins such as the β ribbon. Surprisingly, the data also show no direct correlation between gliding motility and infectivity, since MIC2 and RevCharge mutants show severely reduced motility yet invade almost normally the mosquito and the mouse. This questions the relevance of the gliding assay. How do sporozoites with reduced gliding motility invade mosquito salivary glands and mouse cells?

We agree with the reviewer that there is no clear correlation between the overall surface charge of the I-domains in general. However, the correlation between phenotype and charge becomes more significant in the RevCharge mutant where the charges in the perimeter of the MIDAS motif are altered in an otherwise unchanged TRAP. To avoid that the reader is misled we changed the wording along the manuscript to highlight this difference more clearly.

The RevCharge mutant allowed some striking observations. This mutant shows a strong phenotype in gliding motility in vitro and an uncoupling of salivary gland and mouse infection. The RevCharge sporozoites enter normally into salivary glands but are much less infectious to mice. This highlights that the charge around the MIDAS must be important in the skin and/or liver and less so for salivary gland invasion. However, we refrain from overinterpreting this data towards explaining the results obtained with the other I-domain swaps.

The value (and problem) of the gliding assay on glass is often misunderstood. The striking phenomenon that sporozoites cannot glide in vitro but invade just fine in vivo has most likely to do with the two-dimensionality of the gliding assay as we also discussed in Bane et al., 2016, where the coronin(-) mutant showed a similar phenotype: highly compromised gliding but no difference in skin migration. Also *P. falciparum* sporozoites do not glide in this standard assay used for rodent-infecting *Plasmodium* sporozoites, while their motility in the skin (ie in 3D) is very similar (see recent BioRxiv paper by Photini Sinnis, doi: https://doi.org/10.1101/716878). This, however doesn’t diminish the use of the gliding assay. On the contrary, this simple and highly quantitative assay allows to detect minor but nevertheless interesting phenotypes that will help us to understand how the proteins of the parasite act together to ensure efficient gliding (see e.g. our discussion in Moreau et al., PLoS Pathogens 2017). Please think of the migration assays used for mammalian cells, most of which are similarly simple. They are used to dissect the molecular mechanisms governing cell migration. Often effects of a single gene deletion/mutation can lead to a better understanding of the mechanics while they have little consequence when observed in vivo.

Maybe a simple way to address these issues would be to perform gliding assays not merely on glass slides, as performed here, but on specific substrates, including known ligands of MIC2, aX or aL integrins. This would strengthen the conclusion that the effects of the I-domain modifications are really due to changes in the adhesional properties of the chimeric proteins. For example, do sporozoites expressing the aL-I domain recover gliding motility on ICAM1?

Completely agreed: Please do see our answer to point 2 above. We are currently working on establishing assays that use substrates of different stiffness in both 2D and 3D.

2) The phenotype of the RevCharge mutant should be more thoroughly investigated to better document a defect in sporozoite infectivity for the liver. Figure 6F shows a partial reduction of parasite fitness in the mouse, which could be quantified more precisely. In vitro assays in HepG2 cultures, as performed with the MIC2 mutant, should be included. This is important to strengthen or invalidate the hypothesis that charges in the I-domain play an important role in sporozoite infectivity.

We performed an invasion assay as well as a liver stage assay of the RevCharge mutant compared to the control line TRAP-I. The invasion capacity of the RevCharge mutant is severely impaired while liver stage development seems to be unaffected similar to the results obtained for the MIC2-I mutant. The new data are presented in Figure 6—figure supplement 1 (associated with Figure 6 of the manuscript). We also added more in vivo infection experiments with the RevCharge mutant, which are reported in Tables 2 and 3.

3) The authors checked by immunofluorescence that TRAP is expressed in the mutant sporozoites. However, they should also provide evidence that the level of expression of TRAP is unaffected in the mutants, ideally using Western Blot as shown for the RevCharge mutant. This is particularly important as sequences codon-optimized for *E. coli* were used, which may differentially affect the level of expression of the proteins. Why using *E. coli* optimized sequences?

We included now a western blot of hemolymph derived sporozoites from all chimeras that do not enter salivary glands in sufficient amounts in Figure 3C. We also include a Western blot comparing WT with MIC2 from salivary gland sporozoites (Figure 3E). This shows that TRAP expression is similar to the control (TRAP-I). Please note that we could not detect any phenotypic difference between parasite lines expressing the codon-optimized TRAP (TRAP-I) and wild type sporozoites (compare Figure 2B with Figure 3F and G as well as Figure 2C with 4A and Tables 1 and 2). We mentioned this now at several occasions in the manuscript.

At the time the study was designed only the codon usage of *P. falciparum* was published but not yet implemented in common online tools for codon usage modification. Therefore one possibility would have been to modify the introduced sequences from *Homo sapiens* and *Toxoplasma* manually according to the codon usage of *P. falciparum* leaving the remaining part of the open reading frame unchanged. We decided not to do so because we were unsure how similar the codon usage of *P. falciparum* and *P. berghei* really is. To avoid a change in codon usage from one species to another within a single ORF we codon optimized all chimeric sequences for *E. coli* K12 that we used to optimize genes in previous studies (Moreau et al., PLoS Pathogens 2017, Douglas et al., PLoS Biology 2018) observing no effect on protein expression. However, for some other genes, we noted that codon modification can have dramatic effects on expression levels (e.g. Spreng et al., EMBO J 2019). Hence the western blots are indeed essential.

Reviewer #2:The manuscript by Klug et al. investigates the biochemical and functional properties of I-domains by genetically replacing the I-domain within an important malaria parasite ligand called TRAP with I domains taken from another parasite (T. gondii) ligand, and two from human integrins. Plasmodium parasites lacking either the entire TRAP locus or just the I-domain of TRAP do not move, or invade the vector salivary glands. The authors find that the exogenous I domains from different sources can, to varying degrees, complement TRAP function resulting in the partial rescue of some motility, and ability to invade. The authors notice that the surface charges of TRAP in the rodent parasite P. berghei are cationic and to determine if this is important for function, mutate these residues to acidic amino acids and show again that the functions of TRAP are impaired.The manuscript is very nicely put together with high quality data, well-prepared figures, and the writing is clear. Overall, I found this a rather strange study where the authors have used the phenotyping available in the Plasmodium berghei rodent infection model to investigate general properties of I domains from different proteins and organisms.

Thank you for this summary of our manuscript. We are glad that the reviewer recognizes our study as unique. The rationale of the experiments was to study the function of I-domains of *Plasmodium* in order to discover specific functionality during the complex migrations of the sporozoite with the hope that this could help in defining receptor ligand interactions. We hence made use of those domains found in the distantly related apicomplexan parasite *T. gondii* and the very distant human integrins. Of course studying human integrin domains could be done more easily in mammalian cell culture.

Major pointsThe authors take several precautions to try to ensure the I-domains retain their function when transplanted in the Plasmodium TRAP sequence by taking structural considerations into account and showing that the proteins are detectable using anti-TRAP antibodies in the transgenic parasites. A key point in my mind is whether the human integrin I-domains really do complement the function of the I-domain in TRAP. In the manuscript, the authors show that the I-domain mutant is a null (Figure 2B) and that the human integrin I -domain replacement parasites are very strong hypomorphs (Figures 3F, 4A); however, there does not appear to be a direct side-by-side comparison – e.g. injecting 25,000 HL sporozoites of the TRAP I-domain deletion alongside the integrin replacements in Figure 3—figure supplement 3B, for example. Also, are the authors able to demonstrate that the inserted I-domains retain their functions from their original proteins – is there a known ligand for the inserted integrin I-domains?

We have now added these experiments as described above in our response to the editor.

For the revCharge mutant, the authors imply that the loss of motility and ability to infect mice is due to the alteration in I-domain ligand binding, but could not an alternative explanation be that a proportion of the protein no longer folds correctly or that interactions with other parasite proteins necessary for full function are altered? The authors suggest the role of the I-domain is to interact with the host substrate, but these are not defined in the manuscript making it conceptually a rather large leap to link the mutations made in the protein to the phenotypes seen in the parasite.

We also addressed this question in the response to the editor.

Reviewer #3:In this study the authors swap integrin "I" domains from MIC2 and mammalian integrins with the I-domain of TRAP, the key motility adhesin of malaria sporozoites. Though the domain structure of the extracellular portion of TRAP has long been known to include an integrin-like I-domain, previous mutational studies of this domain were not that informative and the idea to perform domain-swapping experiments between Plasmodium, Toxoplasma and mammalian integrin I-domains is creative and potentially informative. Overall, the MIC2 swapped TRAP and the charge-altered TRAP are the most informative data while the conclusions drawn from the a-L and a-X swapped mutants are not justified by the data shown.

We thank the reviewer for the constructive critique and hope that he/she agrees that the newly added data in Figure 6 and Table 3 improved the conclusions for both RevCharge and human integrin domains.

1) The human integrin I-domain swapped mutants essentially behave like the TRAP KO. There is an elegant explanation in the Discussion as to why the a-L and a-X domains may not open and close in the same way as the I-domains of TRAP and MIC2. However, in the face of no data to support this or another hypothesis, it is equally plausible that there is some significant misfolding on the TRAP with a mammalian I domain. It would be important to show that this is not the case. Since there are commercially available antibodies to human integrins, can they be used, especially ones specific for conformational epitopes, to confirm the overall folding of the a-L and a-X domains.

As described above in our response to reviewer 1, we have added additional infection experiments to Table 2 and the newly added Table 3 that support the folding and functionality of the integrin αX I domain. We did not use the specific antibodies as we feel that for proper controls we would have to generate new parasite lines with mutations in the respective epitopes. We also have preliminary data that conformationally trapped TRAP I-domains influence in vivo and in vitro motility and infectivity to similar degrees as those observed for aX I-domains. However, much more work needs to be done before these experiments will be conclusive and could be published and it would hence go beyond the scope of this manuscript to include those data.

2) The data shown for the a-L and a-X mutants do not support the conclusions. These mutants have less than 200 sporozoites in their salivary glands, the a-L mutant is not motile and not infectious while the a-X mutant is minimally motile and very poorly infectious after IV inoculation. These data look like the TRAP KO (Sultan et al., 1997), where they observed 100 to 200 sporozoites in the salivary glands and had a few infected mice when sporozoites were inoculated IV. Given this, how can they state: "Parasites expressing TRAP with I-domains from other organisms including humans can have nearly intact motility and invasion.” Or "…I-domain of TRAP is required to mediate adhesional properties which can be partially preserved when the native I-domain is replaced by I-domains from human integrins…". These statements are in the Introduction and Abstract but additionally, a central theme of the Discussion is that the a-L and a-X mutants are sufficiently intact to draw meaningful conclusions.

Thanks for this remark, which partly reflects a long discussion we had while putting together the manuscript. Based on the new in vivo results (Tables 2 and 3) we agree with the reviewer that the aL-I domain is most likely nonfunctional and changed the wording across the manuscript accordingly; please also see our comments above.

There are some differences between our study and Sultan et al., 1997 that could explain different results obtained for infectivity and numbers of sporozoites observed in the salivary glands. One explanation is the wild type ANKA background used in our study compared to the NK65 strain used by Sultan et al. Strain-specific differences have been often observed in malaria research and might also explain these subtle differences in salivary gland invasion and in vivo infectivity in absence of TRAP between different strains. Another explanation might be the different levels of infection from one insectary to another. Indeed, while we repeated the infections we obtained higher infection levels and in some of these infections some sporozoites could be seen in the salivary glands (still not as high as in Sultan et al. though). The data is included in Table 2 and we modified our text accordingly.

These lower numbers might also arise from the different design of the TRAP knockout lines. Sultan et al. generated a first knockout by single crossover integration into the coding sequence of TRAP. This design can cause the looping out of the selection marker thus restoring wild type parasites. As a consequence the authors might have observed low levels of salivary gland sporozoites (about 300-400 per infected mosquito). Indeed these sporozoites were able to infect mice and were shown by Southern blotting to be wild type parasites, i.e. they did indeed loop out the integrated sequence. To overcome this problem Sultan et al. created two additional knockout lines by double crossover integration to prevent unwanted recombination events. Unfortunately the homology arms that were chosen to initiate recombination are placed inside the open reading frame leaving the coding sequence for the N- and C-terminus of TRAP intact potentially leading to some activity that could explain why numbers for salivary gland sporozoites increased in both lines about 2-fold (about 700 salivary gland sporozoites per infected mosquito) compared to the single crossover knockout and why sporozoites are still infective if injected in high numbers. Alternatively, this difference is due to the variability of infection as we noticed. The important fact is that the lines show at least 90% reduction in salivary gland infection.

In contrast to this approach the design for the TRAP knockout in our study excludes most of the coding sequence including the start codon to prevent any residual transcriptional activity. We couldn’t observe any productive motility and any infectivity of this line independent if 500k midgut sporozoites or 25k hemolymph sporozoites were injected.

3) TRAP expression in mutant lines is verified by immunofluorescence that is not quantified. Since decreased levels of mutant TRAP could have phenotypes due to the mutation or to expression level, its critical to have western blot verification of mutant TRAP expression compared to WT TRAP expression, in all lines analyzed.

To exclude that differences in protein expression between the different lines are affecting the phenotype we performed western blotting on hemolymph sporozoites of those lines that did not enter the salivary glands. This showed similar TRAP expression in all mutants. The new western blot results were implemented in Figure 3.

[Editors’ note: what follows is the authors’ response to the second round of review.]

Revisions for this paper:One major reservations of the previous iteration of this study was that the conclusions drawn from the parasites in which the TRAP I-domain was replaced by the human a-L and a-X integrin domains were not justified by the data shown and this has not been corrected in the revised manuscript. Indeed, salivary gland sporozoite numbers of the a-L and a-X mutants are in the same range as the TRAPΔI and TRAP KO parasites, with only the prevalence of infection somewhat increased in the a-X parasites. Infectivity studies demonstrate that the a-L parasites are similar to the TRAPΔI and TRAP KO parasites. With numerous careful experiments they that show the a-X parasites are more infectious than the a-L, TRAPΔI and TRAP KO parasites, and this interesting finding suggests that the charge on a-X may enable the mutant TRAP to have some minimal activity. However, their data show that the a-X sporozoites are about 100-fold decreased in their infectivity compared to controls. It has been previously demonstrated that a one day delay in patency is equivalent to a 10-fold decrease in infection. When 10,000 hemolymph sporozoites are inoculated IV, 100% of control mice are infected with a prepatent period of 4 days. With the a-X parasites only 50% of mice are infected and the prepatent period is 6 days. Thus, even if one ignores the fact that half the mice were not infected, the ones that did get infected had a 100-fold lower functional inoculum.Characterization of this significant attenuation of infectivity of the a-X parasites in mosquito and mammalian hosts as "partial restoration" of TRAP function is misleading. Indeed given the phenotype they describe, it is clear that the a-X parasites cannot be naturally transmitted for a single life cycle. Thus, the paper should focus on the interesting phenotypes of the MIC2 and reverse charge sporozoites, and only compare the a-X parasites to the a-L and TRAP KO parasites where there is a small marginal increase in infectivity supporting the importance of charge in the TRAP I-domain.

We now reworded all remaining overstatements accordingly and also corrected one sentence that still overstated the rescue effect of aX.

The following statements in the paper should be deleted, or changed to only include the MIC-2 mutant.Title:The current title implies that human I domains confer some functionality to TRAP which, given that these mutants have salivary gland sporozoite counts in the range of the TRAP KO, and infectivity levels that are 100-fold decreased, is misleading.

We changed the title omitting the mention of human I domains and introducing the word “can” to show that not all I domains rescue, which now reads: Evolutionarily distant I domains can functionally replace the essential ligand-binding domain of Plasmodium TRAP. We also changed the Abstract to make sure that this is reflected. Please see that manuscript file with tracked changes.

Abstract:"…its replacement with I domains from Toxoplasma MIC2 or the human integrin αX-subunit substantially or partially restores these functions".

We completely rewrote the Abstract to put more emphasis. Doing so we removed “or the human integrin aX-subunit“.

"Results with TRAP, MIC2 and aX-I domains show remarkable interchangeability in their ability to engage with ligands in divergent tissues in arthropoa and chordata".

This sentence only appeared in the initial submission and has already been corrected in the revised version. Yet, please note the re-written Abstract, where this statement is not included anymore.

"…results suggest that apicomplexan parasites appropriated I domains in part for their ability to recognize structurally diverse ligands"

We removed “structurally diverse ligands” and modified the sentence/absrtract accordingly.

"…permit TRAP to function as an adhesin with diverse ligands in both vertebrate and arthropod hosts"The ability to engage with ligands is not addressed here, so wording should be revised. Also, the aX-I domain is poorly functional in TRAP, so "remarkable interchangeability" is not correct in this case.

We removed “with diverse ligands”. The expression “remarkable interchangeability” is no longer part of the manuscript.

Introduction:"Parasites expressing TRAP with I domains from other organisms, including humans show partially rescued motility and invasion."

We removed “including humans” and modified the sentence accordingly.

Results:"This could hint that the charge of αX could compensate for the long evolutionary distance between apicomplexa and mammals."

This sentence was completely deleted.

Discussion:"It was therefore remarkable that TRAP function was largely or partially restored by replacement of its I domain by homologues from Toxoplasma gondi and *Homo sapiens* with only 28% and 18% amino acid sequence identity, respectively."

We removed “*Homo sapiens*” and rephrased the sentence accordingly.

"TRAP function was largely or partially restored by replacement of its I domain". This is true for Toxoplasma MIC2 domain but not human integrin I domains.

As before, this sentence was changed accordingly.

"…immunofluorescence showed normal localization in sporozoites, suggesting that they were properly folded".The immunofluorescence data provide no information on the folding of TRAP.

We completely agree with the reviewer and changed “they were properly folded” to “trafficking was not impaired”.

"Infectivity of hemolymph αX-I and αL-I sporozoites in mice was decreased at least 4-fold and 15-fold, respectively." Infectivity in the a-L sporozoites was similar to the TRAP KO, and in the a-X sporozoites was 100-fold decreased compared to controls.

We rephrased the sentence accordingly.

“The findings that I domains from organisms that encounter none of the same ligands as TRAP can replace its function in mediating salivary gland invasion and infection of mice strongly suggest that the I domain functions to provide traction for motility and cell invasion rather than tissue tropism.” This should specify that they are referring to replacement with MIC2.

We specified this sentence.

"…the I domain of TRAP is required for gliding motility and invasion by sporozoites of mosquito salivary glands and mammalian liver, without a requirement for evolved specificity".The data with the human I domains show that any I domain cannot confer function in TRAP, so this statement is not correct.

We do think that this sentence is still correct considering that the common ancestor of Toxoplasma and Plasmodium is about 800 million year old. We did modify by including a “strong” such that it reads “…without a strong requirement…” to modulate our statement.

The authors attempted to analyze the motility of mutant sporozoites on various substrates. They indicate that they "were not able to measure a significant effect of a specific ligand".However, Figure 1 shows that aX-I and aL-I activated on BSA are mostly floating, while on the other substrates sporozoites are mostly attached or display unproductive motility. This could be an indication that the human I-domains retained some adhesive properties. However, it would be useful to have the same type of data with the I domain-deleted mutant, and the threshold, as indicated in A, should also be shown on the other panels.

Thanks for pointing this out. Indeed the differences in the amount of unproductive gliding sporozoites are interesting and could indicate minimal TRAP activity. We are planning to investigate this in more detail making further use of the TRAP mutant library we created. As explained in our last rebuttal, a proper analysis will require a functional linkage of the ligands to a surface. Therefore, we think that now testing of the I domain deletion mutant alone would be not advance our understanding significantly. Also, this would require a repeat of many experiments as the peptides would have to be reordered.

We now include the threshold also in the panel showing gliding motility on different substrates with MIC2-I salivary gland sporozoites. We didn’t include a negative control (RPMI only) for gliding assays with hemolymph sporozoites because these sporozoites are difficult to activate even in the presence of the strong activating compound BSA (only about 20% of wild type hemolymph sporozoites perform active movement while 50-70% of wild type salivary gland sporozoites do active gliding). As a consequence a threshold is not indicated for these experiments.

**References**

Bilsland, C. A. G., Diamond, M. S. and Springer, T. A. (1994) ‘The leukocyte lntegrin pl50,95 (CD11 c/CD18) as a Receptor for iC3b’, *Journal of Immunology (Baltimore, Md : 1950)*, 152, pp. 4582–4589.Gao, C. et al. (2011) ‘Heparin promotes platelet responsiveness by potentiating αIIbβ3-mediated outside-in signaling’, *Blood*. doi: 10.1182/blood-2010-09-307751.